# Protection and consolidation of stone heritage by self-inoculation with indigenous carbonatogenic bacterial communities

Fadwa Jroundi[1], Mara Schiro[2], Encarnación Ruiz-Agudo[2], Kerstin Elert[2], Inés Martín-Sánchez[1], María Teresa González-Muñoz[1] & Carlos Rodriguez-Navarro[2]

Enhanced salt weathering resulting from global warming and increasing environmental pollution is endangering the survival of stone monuments and artworks. To mitigate the effects of these deleterious processes, numerous conservation treatments have been applied that, however, show limited efficacy. Here we present a novel, environmentally friendly, bacterial self-inoculation approach for the conservation of stone, based on the isolation of an indigenous community of carbonatogenic bacteria from salt damaged stone, followed by their culture and re-application back onto the same stone. This method results in an effective consolidation and protection due to the formation of an abundant and exceptionally strong hybrid cement consisting of nanostructured bacterial $CaCO_3$ and bacterially derived organics, and the passivating effect of bacterial exopolymeric substances (EPS) covering the substrate. The fact that the isolated and identified bacterial community is common to many stone artworks may enable worldwide application of this novel conservation methodology.

[1] Department of Microbiology, Faculty of Science, University of Granada, Avda. Fuentenueva s/n, 18071 Granada, Spain. [2] Department of Mineralogy and Petrology, Faculty of Science, University of Granada, Avda. Fuentenueva s/n, 18071 Granada, Spain. Correspondence and requests for materials should be addressed to C.R-N. (email: carlosrn@ugr.es)

Landmarks of the world's cultural heritage, like the pyramids and the Sphinx in Egypt, the cathedrals in Europe and America, and the Maya temples in Mesoamerica, built of or carved out of stone, are inexorably crumbling due to physical, chemical and biological weathering[1–3]. This gradual and irreversible deterioration has accelerated over the last century and is expected to continue at an even higher rate in the near future, due to increasing air pollution[4] and enhanced salt damage associated with climate change[5]. The conservation of such historic and culturally significant stone artworks typically involves the in situ protection and/or consolidation of damaged stones, commonly by applying organic polymers, alkoxysilanes, or inorganic materials, which bind loose grains and/or fill cracks[2, 6]. However, these conventional conservation methods have limitations and disadvantages, including physical-chemical incompatibility (e.g., organic polymers), alteration of the stone appearance, and the formation of superficial films that induce further damage by pore blocking that limits water vapor transport[2, 7]. Moreover, some of these conventional treatments are not environmentally friendly because they release toxic chemicals, and the treatments themselves can undergo degradation (e.g., biodeterioration of organic polymers)[2].

In recent years, bacterial biomineralization, also called bacterial carbonatogenesis, has emerged as an environmentally friendly methodology for the conservation of decayed stones, particularly those made up of carbonate minerals (such as limestone and marble, which are among the most commonly used in artworks and monuments)[8–13]. This conservation strategy takes advantage of bacterially induced calcium carbonate precipitation to cement carbonate rocks, a phenomenon that is widespread in natural environments such as soils, caves, lakes, and oceans[14–16]. Two main strategies for decayed stone conservation that have been previously presented[3], are also tested and presented here. The first consists of the application to weathered stone of a nutritive solution along with exogenous[8–13, 17–19] or stone-isolated[20] single bacterial cultures with proven carbonatogenic ability. The second consists of the application of a sterile nutritive solution that aims to activate the carbonatogenic bacteria from amongst the microbial community of the stone[3, 18, 19, 21–23]. Both strategies result in the formation of calcium carbonate cements that protect and consolidate decayed stones.

A limitation of the first strategy is that its effectiveness depends on characteristics of the stone substrate (e.g., porosity, mineralogy, level of degradation), the presence of soluble salts, the type of microorganism and bacterial load applied, and its interaction with indigenous microbiota[3, 12, 13]. Note that exogenous bacteria or stone-isolated single bacterial cultures are likely at a competitive disadvantage with respect to the microorganisms already present in equilibrium and/or better adapted to the local microenvironment in the stone. Moreover, the application of an exogenous inoculum or a stone-isolated single bacterial culture may alter the dynamics of the bacterial community in the stone in an unpredicted way[19, 23]. Most importantly, we show here that both strategies have limited efficacy when applied to heavily salt weathered stone. Salt weathering can be highly deleterious[24–27] and usually results in at least partial filling of pore spaces with soluble salts, which upon the addition of water vapor or water, deliquesce or dissolve to form brines. These crystalline salts or brines may inhibit the proliferation of exogenous or indigenous carbonatogenic bacteria.

In order to bypass the limitations associated with the two bacterial conservation strategies described above, a new self-inoculation method is presented and studied here. An indigenous bacterial community was recovered and activated from carbonate stone in a historic building (San Jeronimo Monastery, Granada, Spain) that had suffered substantial damage due to salt weathering. In the laboratory, bacterial strains among the bacterial community were identified using molecular methods, including genomic DNA extraction followed by REP-PCR clustering and 16S rRNA gene amplification and sequencing. The carbonatogenic capacity of the bacterial community was tested following activation with a patented nutritive solution (M-3P), which is designed to specifically activate the carbonatogenic bacteria from among the community of bacteria present in a stone[22]. The whole carbonatogenic bacterial community was then applied in situ (i.e., self-inoculation) along with M-3P to the same stone from which it was isolated. Additionally, it was applied in the laboratory on sterile and unweathered calcite substrates for the sake of evaluating the time evolution of the formation and dissolution of the treatment layer. The protection and consolidation effectiveness of this novel conservation method was evaluated both in situ and in

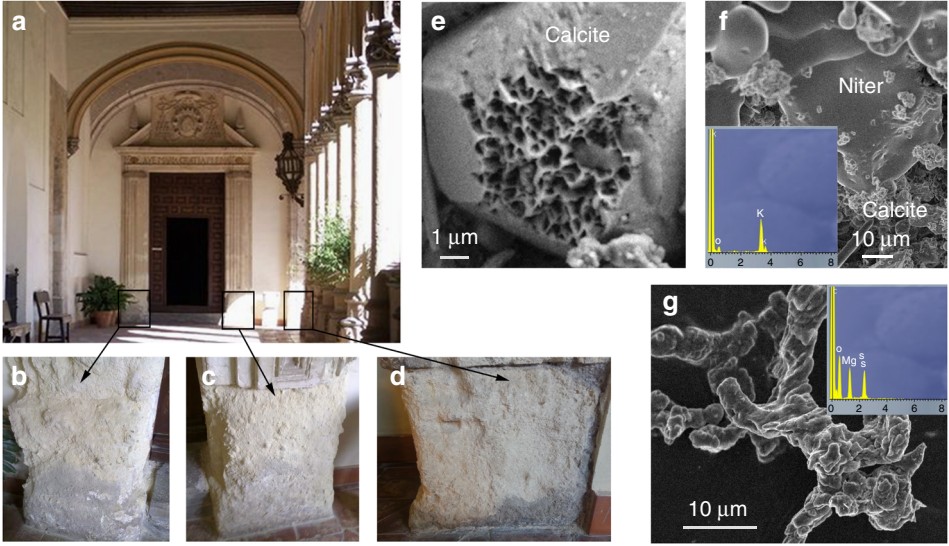

**Fig. 1** Stone damage at the cloister entrance of San Jeronimo Monastery. **a** General overview. Detail of the stone blocks used for the bio-consolidation treatments: *Myxococcus xanthus* treatment (**b**), M-3P nutritive solution treatment (**c**), self-inoculation bio-treatment (**d**). Scanning electron microscopy (SEM) photomicrographs of corroded calcite (**e**), niter (KNO$_3$) (EDS spectrum in inset) (**f**) and hexahydrite (MgSO$_4$·6H$_2$O) salt crystals (**g**) in the weathered calcarenite stone (EDS spectrum in inset)

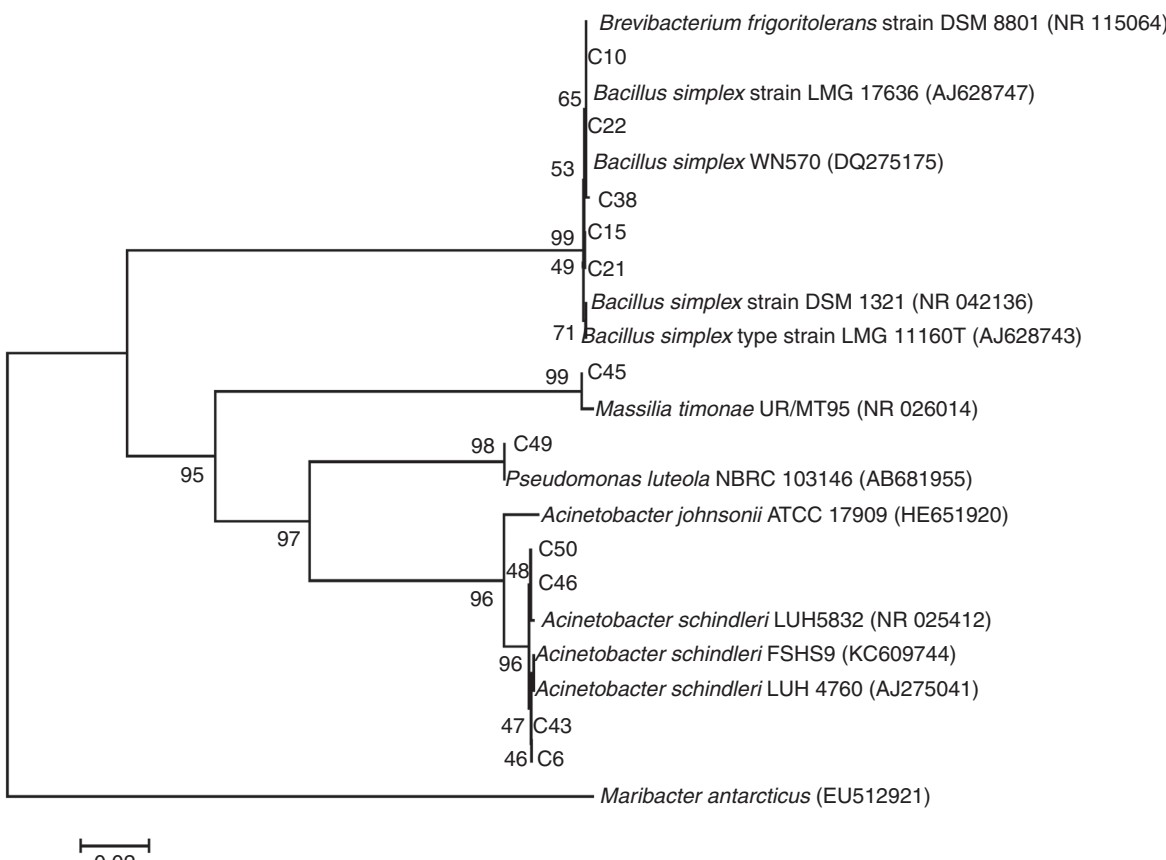

**Fig. 2** Phylogenetic tree of the bacterial community isolated from decayed calcarenite stones in San Jeronimo Monastery. The phylogenetic tree shows the taxonomic location of a representative strain from each REP-PCR group based on 16S rRNA sequences. Reference strains are also indicated. The significance of each branch is indicated by bootstrap values calculated for 1000 subsets. *Scale bar*: two nucleotides substitutions per 100 nucleotides

the laboratory using a range of analytical techniques. These results demonstrate that this self-inoculation treatment provides greater protection and consolidation effectiveness for salt weathered stones than the previously proposed bio-treatments tested here.

## Results

**Characterization of stone samples**. The highly decayed porous bioclastic calcarenite stone of the San Jeronimo Monastery (Granada, Spain) studied here shows extensive granular disintegration (sanding), surface pitting, and a general loss of surface relief due to both salt-enhanced calcite dissolution (Fig. 1), and salt-induced physical weathering resulting from the crystallization of (in order of decreasing abundance) syngenite ($K_2Ca(SO_4)_2 \cdot H_2O$), niter ($KNO_3$), hexahydrite ($MgSO_4 \cdot 6H_2O$), gypsum ($CaSO_4 \cdot 2H_2O$), and halite (NaCl) (Fig. 1f, g and Supplementary Fig. 1). Drilling resistance (DR) of the unweathered stone blocks displayed a relatively constant value of $2.6 \pm 0.8$ N along a 10 mm depth profile (Supplementary Fig. 2). In contrast, weathered stone blocks displayed reduced DR values ($\sim 1.8 \pm 0.9$ N) in the first ~ 3 mm of the depth profile (Supplementary Fig. 2). This is due to cement dissolution and granular disintegration/decohesion associated with salt weathering, a process that is typically concentrated within the first few mm below the stone surface[25].

**Identification of the bacterial community used as inoculant**. Fifty bacterial isolates were recovered from the stone blocks selected from San Jeronimo Monastery (Fig. 1b–d), which clustered into 14 different REP-groups that were identified as seven different species belonging to three phylogenetic groups (Fig. 2):

Firmicutes was dominant (~ 79%) within the culturable bacterial community, followed by ~16% of Gamma-proteobacteria, and ~5% of Beta-proteobacteria (Table 1). Among the Firmicutes, four species were distinguished and showed 99% similarity with sequences of *Brevibacterium frigoritolerans* (the most abundant species), *Bacillus simplex*, and *Bacillus thuringiensis* (the less abundant species), and 100% similarity with *Bacillus* sp. Among the Beta-proteobacteria, only one species was detected and was related with 99% similarity to *Massilia timonae*. Three species from the Gamma-proteobacteria were identified; they related to *Acinetobacter schindleri*, *Acinetobacter johnsonii*, and *Pseudomonas luteola* with 99, 98, and 100% similarity, respectively, with the first one being the most abundant.

**Carbonatogenic capacity of the bacterial community**. Microscopic observations of M-3P solid medium plates 48 h after inoculation with bacterial isolates show a high $CaCO_3$ production in all cases (i.e., every single bacterial isolate cultured in M-3P medium), both within the colonies and in the solid culture medium (Supplementary Fig. 3a–c). XRD analysis identified these bacterial precipitates as abundant calcite and < 10 wt% of vaterite (Supplementary Fig. 3d), which continuously grew in size with increased incubation time.

**The early stages of bacterial community $CaCO_3$ mineralization**. To gain an insight into the mechanism of bacterial $CaCO_3$ formation, the whole community of isolated carbonatogenic bacteria were inoculated in M-3P liquid medium and aliquots were collected at predetermined time intervals. Transmission electron microscopy (TEM) analysis of the initial stages of bacterial

**Table 1 Phylogenetic classification of the bacterial community activated with the M-3P nutritive solution and used as inoculant for the bio-consolidation treatment in San Jeronimo Monastery**

| Rep-PCR groups | Name of isolates* | Closest identified phylogenetic relatives | Similarity (%) | Accession number of the closest strains | Phylum/Order | Abundance (%) |
|---|---|---|---|---|---|---|
| I | **C6**-C20 | *Acinetobacter schindleri* | 99 | AJ275041 | Gamma-proteobacteria | 5.26 |
| II | **C43** | *Acinetobacter schindleri* | 99 | KC609744 | | 2.63 |
| III | **C46** | *Acinetobacter schindleri* | 99 | NR_025412 | | 2.63 |
| IV | **C49** | *Pseudomonas luteola* | 100 | AB681955 | | 2.63 |
| V | **C50** | *Acinetobacter johnsonii* | 98 | HE651920 | | 2.63 |
| VI | **C45**-C47 | *Massilia timonae* | 99 | NR_026014 | Beta-proteobacteria | 5.26 |
| VII | **C11**-C17 | *Bacillus thuringiensis* | 99 | AM747222 | Firmicutes | 5.26 |
| VIII | **C34**-C41-C44 | *Bacillus thuringiensis* | 99 | KJ496385 | | 7.9 |
| IX | **C22**-C13 | *Bacillus simplex* | 99 | DQ275175 | | 5.26 |
| X | **C21** | *Bacillus simplex* | 99 | AJ628743 | | 2.63 |
| XI | **C38**-C40 | *Bacillus simplex* | 99 | AJ628747 | | 5.26 |
| XII | **C15** | *Bacillus simplex* | 99 | NR_042136 | | 2.63 |
| XIII | **C16** | *Bacillus* sp. | 100 | AJ315057 | | 2.63 |
| XIV | **C10**-C18-C19-C25-C26-C27-C28-C29-C30-C32-C33-C36-C37-C39- C24-C23-C31-C35 | *Brevibacterium frigoritolerans* | 99 | NR_115064 | | 47.37 |

*Strains in bold are the representatives of each REP-PCR group

$CaCO_3$ mineralization (17 h after inoculation) showed an accumulation of nanometer-sized precipitates onto the bacterial cell walls (Fig. 3a). These precipitates were identified as amorphous calcium carbonate (ACC) based on their selected area electron diffraction (SAED) pattern (Fig. 3b), which showed diffuse rings. Over time (up to 48 h) larger porous aggregates (~ 100–500 nm) made up of nanoparticles (~ 10–30 nm in size) interspersed by low electron-absorbing material (organics) were observed attached to the bacterial cell walls (Figs. 3c and e). They typically displayed an overall rhombohedral shape (Fig. 3e), with SAED analysis showing that such nanostructures were single-crystalline calcite (Fig. 3d and inset in Fig. 3e). Their textural and crystallographic features are reminiscent of calcite mesocrystals[28]. At longer incubation times ($\geq 48$ h) these $CaCO_3$ structures grew in size and the individual bacterial cells could no longer be identified, likely because they were entombed within the carbonate structures (Supplementary Fig. 4).

To evaluate the potential role of the calcitic stone substrate on bacterial $CaCO_3$ formation, the isolated and activated carbonatogenic bacterial community was inoculated onto sterile calcite single crystals in M-3P liquid medium.

Field emission scanning electron microscopy (FESEM) and transmission scanning electron microscopy (STEM) showed that during the first 20 h, bacteria formed dense colonies partially covering the calcite substrate (Fig. 4a and b). At this point a few scattered newly formed $CaCO_3$ structures were observed associated with the bacterial cells as well as bacterial extracellular polymeric substances (EPS; Fig. 4c). Some individual calcified bacterial cells were also observed (Fig. 4d). Figure 4e shows a STEM detail of a bacterial cell partially entombed into a rhombohedral calcite crystal. Micro-Raman spectroscopy analysis confirmed that such crystals were calcite (Supplementary Fig. 5). After 24 h, the partial calcification of EPS was observed (Fig. 4f). Scattered and oriented micrometer-sized rhombohedral-shaped and/or spheroidal calcite structures entombed attached bacterial cells, and formed on the original calcite surface with their growth controlled by the (104)$_{calcite}$ substrate (i.e., self-epitaxy; Fig. 4g), as has been previously reported[13]. Over time (>48 h) these calcitic structures grew in height and fully covered the calcite substrate surface (Fig. 4h). These larger semi-spherical or shapeless $CaCO_3$ structures were formed by aggregates of nanoparticles (~ 30–50

nm in size) embedded in EPS (Fig. 4i). Overall these observations indicate that the isolated indigenous carbonatogenic bacterial community was highly effective in the formation of abundant $CaCO_3$ (calcite).

**Evaluation of in situ treatment effectiveness**. The effectiveness of the new self-inoculation bio-treatment was tested in situ at the San Jeronimo Monastery. For comparison purposes, the *Myxococcus xanthus* treatment and the sterile M-3P treatment were also performed. All three treatments were applied on adjacent stone blocks with similar exposure and decay levels (Fig. 1).

In the case of the *M. xanthus* and the M-3P treatments, the in situ treatments resulted in limited consolidation, which was mostly lost within a relatively short time-span. Despite data scattering (likely due to the heterogeneous nature of the calcarenite substrate), the peeling tape test indicated that there was a slight surface consolidation 5 months after the treatments that had disappeared in both treatment areas after 12 months (Fig. 5a). Drilling resistance (DR) values obtained 24 months after treatment were slightly higher along the first ~ 5 mm of the depth profile than in unconsolidated, yet unweathered, deeper areas (from 5 to 10 mm deep), especially in the case of the sterile M-3P treatment (Fig. 5b and c). SEM observations of stones sampled 24 months after treatment showed negligible amounts of bacterial calcium carbonate cement and EPS, the lack of which likely explains the limited consolidation. Chromatic changes ($\Delta E$ values; Supplementary Table 1) over the 24 month study period were $\leq 3.6$. Moreover, $t$-tests showed that $p$-values were higher than 0.05, with the exception of the stone treated with M-3P (after 12 months) where the $p$-value was slightly below 0.05 (Supplementary Table 1). These results show that in nearly all cases the null hypothesis, namely that the difference between the mean $\Delta E$ values of the control and treated stones is zero, could not be rejected. This means that color changes were not statistically significant (95% confidence interval) in most cases, and in the only case where color changes were statistically significant, the $\Delta E$ value was still below the generally acceptable maximum $\Delta E = 5$[29].

In the case of the self-inoculation bio-treatment, the peeling tape test showed vastly superior surface consolidation compared

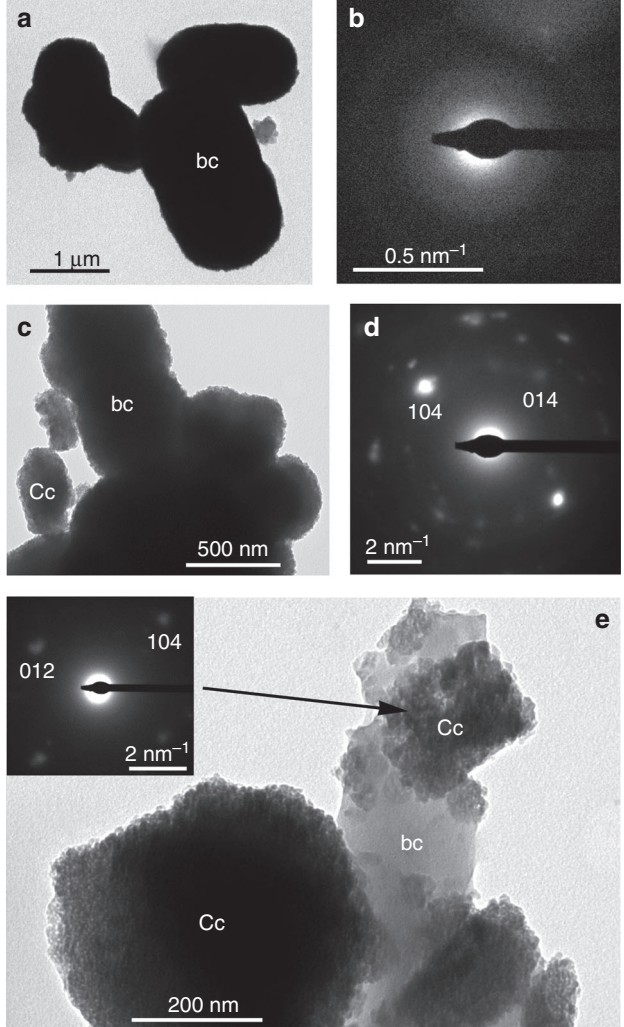

**Fig. 3** TEM analysis of bacterial calcium carbonate precipitates. **a** After 17 h incubation time bacterial cells (bc) are covered by nanoparticle precipitates; **b** SAED pattern of the structures in **a** showing diffuse haloes demonstrating the amorphous nature of such precipitates; **c** after 48 h incubation time, larger precipitates made up of nanoparticles are observed around the bacterial cells; **d** the $[\bar{4}41]$-zone axis SAED pattern of precipitates in **c** shows they are single-crystalline calcite (Cc); **e** detail of calcite precipitates ($[\bar{4}2\bar{1}]$-zone axis SAED pattern in inset) formed around bacterial cells. Note their nanogranular porous structure

to both the untreated stone or the other two treatment methods (Fig. 5a), and the increased surface consolidation was sustained over the duration of the study period (24 months). Similarly, DR measurements performed 24 months after the self-inoculation treatment showed a significant increase in stone cohesion as compared to the untreated stone (Fig. 5d). Such a strengthening was most marked in the first ~ 3–5 mm along the depth profile, where a maximum DR value of $10 \pm 6$ N was achieved, which is nearly 4 times higher than the DR value of the unweathered and untreated stone ($2.6 \pm 0.8$ N; Supplementary Fig. 2). While such an increase in drilling resistance might appear excessive and could potentially lead to detrimental differences in mechanical behavior between the consolidated part and the unweathered stone, the fact that 2 years after the treatment application no scaling or surface material loss was observed (see peeling tape test results, above), corroborates that the strengthening achieved was pivotal for the successful consolidation of the stone. Further

beneath the surface (>~ 5 mm), no further bio-consolidation was achieved as shown by DR values that approached a nearly constant value of $2.3 \pm 0.9$ N. Note that average DR values were very similar, within error, across all of the samples at depths between 5 and 10 mm. In addition, the unweathered and untreated stone blocks displayed relatively consistent DR across all depths (Supplementary Fig. 2). Monitoring of color changes after treatment indicated that no significant changes were observed, the $\Delta E$ value being $3.8 \pm 1.7$ (Supplementary Table 1). In this case, $t$-tests showed that $p$-values were higher than 0.05, with the exception of color changes after 24 months, where the $p$-value was below 0.05 (Supplementary Table 1). Nonetheless, even in this latter case where the color change was statistically significant, the $\Delta E$ value was below 5[29].

The strengthening of the stones subjected to the self-inoculation bio-treatment is attributed to the formation of abundant calcium carbonate in the form of µm-size aggregates of ~ 30–100 nm-size nanocrystals embedded in EPS (Fig. 6a–e). Such newly formed bacterial $CaCO_3$ cemented the stone without pore blocking (Fig. 6b) and displayed similar nanogranular features to those present in the bacterial calcite grown in the laboratory on calcite single crystals (compare Figs 4i and 6c). The lack of pore blocking was confirmed by mercury intrusion porosimetry (MIP), which showed a limited reduction in porosity after the treatment (from $27 \pm 1\%$ to $25 \pm 1\%$) with minimal changes in pore size distribution (Supplementary Fig. 6). The overall ~ 10% reduction in porosity further confirms that abundant bacterial calcium carbonate was formed. The identification of the new bacterial $CaCO_3$ cement was facilitated by its significant textural differences compared to the calcite grains in the untreated weathered calcarenite (micrometer-sized rhombohedral crystals -corroded by salt-enhanced dissolution-, see Fig. 1e and 6a). The bacterial origin of this newly formed cement was further corroborated by the presence of calcified bacterial cells (Fig. 6e). XRD analyses of samples removed from treated areas showed that calcite was the only detectable calcium carbonate phase present, both before and after the self-inoculation bio-treatment (Fig. 6f), which is consistent with our laboratory tests (see above sections on Carbonatogenic capacity of the bacterial community and The early stages of bacterial community $CaCO_3$ mineralization) and previous studies showing that bacterially induced calcium carbonate systematically forms as calcite on calcitic substrates and as vaterite on non-calcitic substrates[13]. The formation of abundant bacterial calcite biocement, along with abundant EPS, is related to a high bacterial activity and contributed to the large and long-lasting increase in resistance to salt weathering of the treated stone. Indeed, no granular disintegration and associated stone material loss was observed 24 months after the treatment.

To help link the amount of bacteria in the stone with the observed effectiveness of the treatments, the total bacterial load was monitored before and after treatment. After the M-3P treatment, the total number of bacteria remained very similar (i.e., ~ $10^3$ colony forming units (CFU) $g^{-1}$) before and after the treatment application. The *M. xanthus* treatment led to a decrease in the bacterial population from $10^3$ to 10 CFU $g^{-1}$ after 5 months. In this particular case, no bacteria could be isolated from the stone at 12 and 24 months. Note that *M. xanthus* typically disappears a few days after treatment application[23]. The drastic reduction in bacterial population after treatment may be explained by the competition between this exogenous bacteria and the bacterial community present in the stone, and the low bacterial population may explain the limited consolidation efficacy of this treatment. In both these cases, no deleterious microorganisms (nitrifying bacteria, sulphur-oxidizers, or fungi) were detected on the stones after treatment.

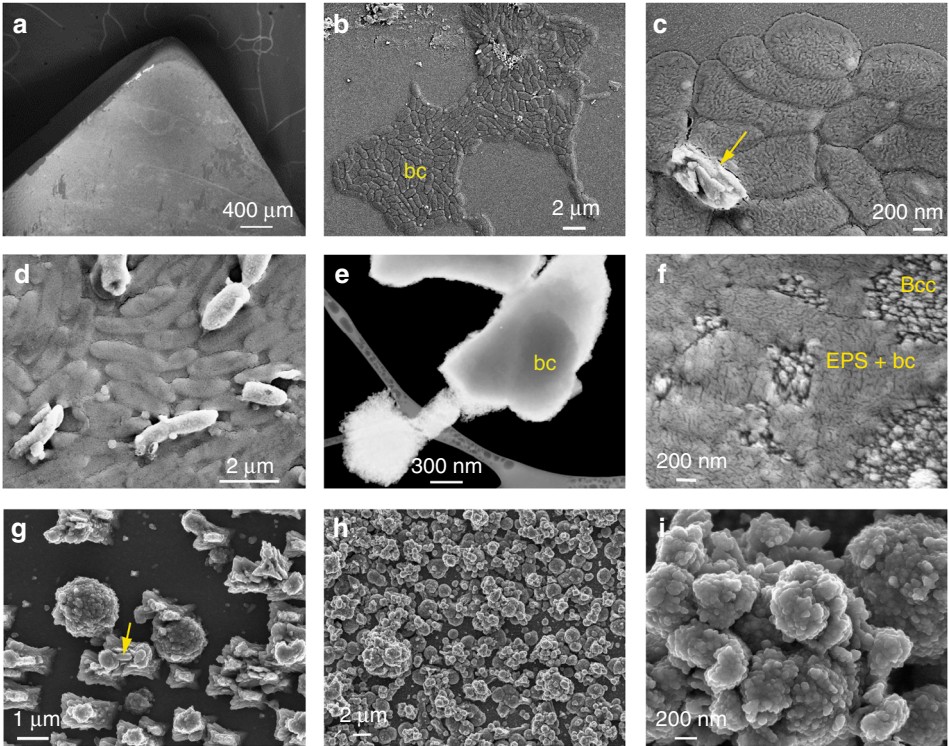

**Fig. 4** SEM photomicrographs of bacterial attachment and CaCO$_3$ biomineralization on calcite. **a** General overview of the calcite substrate subjected to bacterial biomineralization; **b** bacterial cells (bc) onto calcite 17 h after inoculation; **c** detail of bacterial cells and newly formed CaCO$_3$ (*arrow*); **d** initial calcification of bacterial cells that keep their rod-shaped morphology, whereas uncalcified cells (*darker contrast*) collapse during sample preparation (drying) and observation (high vacuum); **e** bright field (negative) SEM image taken with an STEM detector of bacterial calcite crystals formed after 24 h incubation time, one of them including an entombed bacterial cell (bc); **f** bacterially induced CaCO$_3$ crystals (Bcc) along with EPS and collapsed bacterial cells (EPS + bc) on the calcite substrate; **g** bacterial calcite structures formed on the calcite substrate during the early stages of biomineralization (24 h after inoculation). Rod-like bacterial cells are seen partially entombed in newly formed CaCO$_3$ (*arrow*); **h** calcite substrate fully covered by bacterial calcite after 48 h; **i** detail of the nanostructured bacterial calcium carbonate structures shown in **h**

Five months after the self-inoculation bio-treatment, the total number of culturable bacteria found on the stone increased to $1.32 \times 10^7$ CFU g$^{-1}$ from $2.81 \times 10^4$ CFU g$^{-1}$ before treatment. After 24 months it had dropped back down to $2.78 \times 10^4$ CFU g$^{-1}$, close to the pre-treatment value. Remarkably, the percentage of culturable bacteria capable of producing acids in the presence of fermentable carbon decreased from 100% before the treatment to 25%, 24 months after the self-inoculation bio-treatment. Neither nitrifying bacteria nor sulphur-oxidizers were detected after this treatment. The reduction in the population of acid-producing bacteria following treatment is likely due to the fact that they are at a competitive disadvantage compared with activated carbonatogenic bacteria[18]. This situation is favored by the fact that acid-producing bacteria typically use carbohydrates as an energy source, and our nutritive M-3P medium lacks carbohydrates. The total number of fungi showed an increase to $1.88 \times 10^2$ CFU g$^{-1}$ 5 months after treatment from 1.5 CFU g$^{-1}$ before the treatment, but later the fungi disappeared entirely.

**Laboratory evaluation of treatment protective efficacy.** Salt damage, one of the most deleterious weathering phenomena affecting stone artworks and monuments[24], is mainly due to mechanical stress resulting from in-pore salt crystallization[26, 27]. Hence, the observed increase in mechanical strength of in situ bio-treated stones would be effective to protect the stone against such a physical weathering phenomenon, as demonstrated by the peeling tape test and the DR results of the stone block subjected to the self-inoculation bio-treatment (see above). However, saline

solutions, especially those including Mg, enhance calcium carbonate dissolution[30]. We have observed that salt-induced chemical weathering contributes to the overall stone damage at San Jeronimo Monastery (i.e., pervasive dissolution pits on calcite grains, Fig. 1e). To test whether the bacterial treatment could also be effective in reducing salt-induced chemical weathering (i.e., protective effect), we performed nanoscale in situ AFM analyses of the dissolution of calcite crystals after the self-inoculation bio-treatment using the carbonatogenic bacterial community isolated from San Jeronimo Monastery stone blocks. Figure 7 shows time-resolved AFM images of the dissolution of both control (untreated) and bio-treated calcite crystals in contact with 1 M MgSO$_4$ solution. Right after solution injection in the AFM fluid cell, the control (untreated substrate) displayed a high density of rhombohedral dissolution pits formed on the initially flat (10$\bar{1}$4) surface that rapidly deepened and expanded via macrostep retreat along the $\langle \bar{4}41 \rangle$ directions (Fig. 7a–c). In contrast, the bio-treated (10$\bar{1}$4)$_{calcite}$ surface displayed the rough topography of the bacterial calcite overgrowth (i.e., entombed calcified bacterial cells)[13] and experienced negligible dissolution over the course of the dissolution experiment (Fig. 7d–f). Control tests using deionized water showed similar results, the only difference being that the density of dissolution pits on the untreated calcite surface and the rate of step retreat were much lower, whereas the bacterially treated surface experienced no detectable dissolution effects. Similar effects were observed using other saline solutions (e.g., MgCl$_2$ and Na$_2$SO$_4$ solutions of similar ionic strength). Macro-scale dissolution tests (Fig. 8), which spanned 1 week, confirmed that the dissolution rate of calcite crystals was significantly

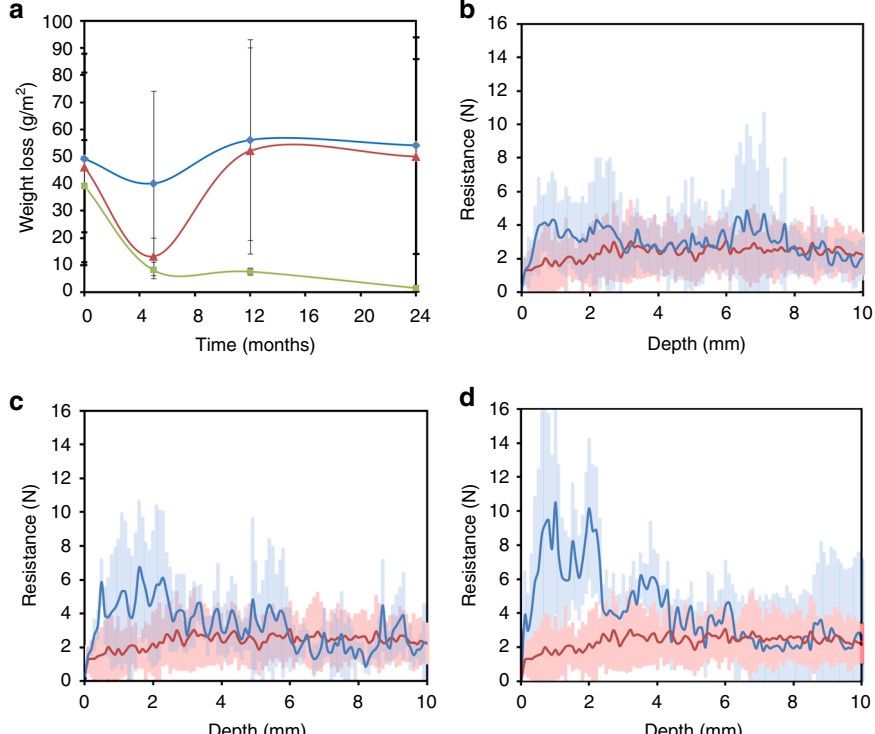

**Fig. 5** Consolidation effect of the in situ bacterial bio-treatments. **a** Peeling tape test (legend: Mx-treatment, *blue line* and *diamond* symbols; M-3P treatment, *red line* and *triangle* symbols; Bacterial community self-inoculation bio-treatment, *green line* and *square* symbols). *Error bars* show s.d. (±1σ). Drilling resistance (DR) along a 1 cm depth profile following Mx-treatment (**b**) (average of at least six measurements per treatment; *blue line*), M-3P treatment (**c**) (average of at least six measurements per treatment; *blue line*), and bacterial community self-inoculation bio-treatment (**d**) (average of at least 6 measurements per treatment; *blue line*). In **b**–**d** the DR of untreated weathered calcarenite stone (average of 14 measurements; *red line*) is presented for comparison. *Shaded areas* in DR graphs show s.d. (±1σ)

increased by the 1 M $MgSO_4$ saline solution compared to $H_2O$. They also showed that the dissolution rate of calcite was strongly reduced after the bacterial community treatment. The differences (reduction) in dissolution rate after bacterial treatment were statistically significant as demonstrated by t-tests showing p-values < 0.05 (0.00005 and 0.000002 for $H_2O$ and 1 M $MgSO_4$ dissolution experiments, respectively). Overall, these results show that the newly formed bacterial calcite is less prone to dissolution both in water and in saline solutions, demonstrating, for the first time, that this self-inoculation bio-treatment produces an effective protective coating against chemical weathering.

## Discussion

The successful consolidation of severely salt damaged stone using the new self-inoculation approach showed that an indigenous community of carbonatogenic bacteria that are well adapted to the harsh, highly saline environment, was the most successful of the tested bio-treatments in terms of stone strengthening throughout the tested time period (up to 24 months) as demonstrated by the peeling tape and DR tests results, and produced no significant color changes or pore plugging. This was achieved by inducing the formation of an abundant amount of exceptionally strong $CaCO_3$ cement that incorporated bacterially derived organics. In addition, no deleterious microbiota was observed at any time after the treatment. Moreover, the treatment resulted in a reduction of potentially deleterious acid-producing microbiota. This is consistent with previous studies showing that an increase in the amount of carbonatogenic bacteria prevents the growth of autochthonous acidifying microbial consortia[18], and demonstrates that the treatment is effective against

biodeterioration. In contrast, the efficacy of the two conventional bacterial consolidation strategies tested here (the M. xanthus and M-3P treatments) that were successfully used in situ in the past to consolidate less damaged calcarenite stones[3, 18, 19], were ineffective in the case of this extremely salt weathered calcarenite stone (Fig. 1), and in the presence of highly saline solutions present in these porous stone blocks. These results show that the laboratory activation of the isolated carbonatogenic bacterial community cultured for 24 h in the M-3P nutritive solution (i.e., at the final stage of the exponential growth) prior to the in situ application (i.e., self-inoculation) of the treatment solution was critical to the effectiveness of this treatment. Right from the beginning of the treatment application, a large number of carbonatogenic bacteria were able to contribute (via their metabolic activity) to induce calcium carbonate precipitation. Note that the effectiveness of any bacterial conservation treatment via calcium carbonate precipitation is directly related to the carbonatogenic bacterial cell load[12].

Prior to the bacterial community self-inoculation treatment it was necessary to first identify the culturable bacteria and evaluate their carbonatogenic capacity. The members of the activated stone-bacterial community were identified as microorganisms commonly found on calcareous stone, as well as in natural carbonate rock outcrops. For instance, the identified Bacillus species are known to play an important role in carbonate deposition in natural habitats[14], including saline soils[31], and have been used for limestone consolidation[11, 12]. Pseudomonas species have been detected, along with Bacillus species, as part of the microbiota in caves[32], on stone monuments[18, 21, 33, 34], and wall paintings, including Paleolithic cave paintings[35]. Brevibacterium, the most abundant bacteria identified here, is known to be able to

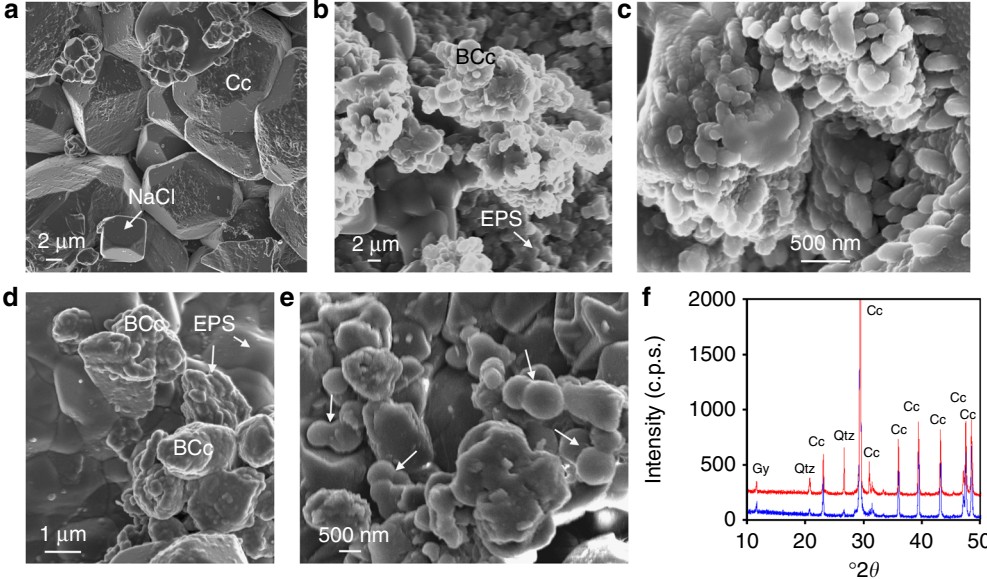

**Fig. 6** Analysis of in situ bio-treated calcarenite stone. SEM images of untreated calcarenite (**a**) and calcarenite treated with the bacterial community (**b**–**e**). **a** Micrometer-sized calcite (Cc) crystals in the control stone show dissolution pits and are covered by salt (e.g., NaCl) crystals; **b** bacterial calcite (BCc) cement shows nanogranular structure surrounded by EPS; **c** detail of the nanogranular structure of calcite biocement interspersed with EPS; **d** bacterial calcite and abundant EPS covering the calcarenite substrate; **e** detail of mineralized bacterial cells (*arrows*); **f** XRD patterns of calcarenite before (*blue*) and after (*red*) bacterial bio-treatment. Cc, calcite; Gy, gypsum; Qtz, quartz

withstand the severe conditions prevailing on stone surfaces[32, 36] and, therefore, frequently predominate over more delicate gram-negative bacteria[37]. *Acinetobacter* sp. is a common microorganism in Mediterranean calcareous stones[21], and shows a high capacity to induce calcium carbonate precipitation on decayed stone[18]. The bacteria isolated and identified here were demonstrated to be halotolerant by their survival in great numbers in the calcarenite stones tested here, which is consistent with previous studies indicating that a significant population of bacteria (especially *Bacillus* spp.) isolated from stones are halotolerant[38].

Our results show that all of the above-mentioned bacteria present in the stone-isolated bacterial community display a high carbonatogenic capacity. These heterotrophic bacteria are able to produce $CO_2$ and $NH_3$ following oxidative deamination of amino acids present in the M-3P nutritive solution (i.e., Bacto Casitone, a hydrolyzed derivative of casein)[10]. As a result, they contribute to an alkalinization of the medium, shifting the $HCO_3^- = CO_3^{2-} + H^+$ equilibrium toward the right. In the presence of $Ca^{2+}$ (present in the M-3P medium as $Ca(CH_3COO)_2 \cdot 4H_2O$) precipitation occurs via the reaction $Ca^{2+} + CO_3^{2-} = CaCO_3$. Precipitation, which preferentially took place on the bacterial cell surface and on EPS, likely occurred in a microenvironment highly supersaturated with respect to $CaCO_3$ (i.e., bacterial biofilm)[39, 40]. The formation of ACC prior to the development of crystalline $CaCO_3$ is consistent with such a high supersaturation[41]. Our results also demonstrate that bacterial $CaCO_3$ mineralization is kinetically-controlled and follows the Ostwald´s step rule, with metastable amorphous phases (ACC) preceding the formation of stable calcite[41]. Formation of precipitates surrounding the bacterial cells (SEM and TEM results) takes place until cell entombment[12]. Transient pores must be present in the newly formed carbonate structures surrounding the bacterial cells to enable transport of nutrients/ions and metabolic by-products (e.g., EPS, $NH_3$ and $CO_2$). However, the eventual sealing of such small pores during entombment would stop the biomineralization process due to cell death[12]. This is consistent with our STEM observations (Fig. 4e) showing entombed bacterial cells surrounded by pore-free calcite/EPS and our MIP results

(Supplementary Fig. 6) showing a reduction, rather than an increase, in the smaller pores <1 μm.

Concerning the protective effects of the new bio-treatment against chemical weathering, in situ AFM and macroscale flow-through dissolution tests unambiguously demonstrate that the treatment drastically reduces the calcite dissolution rates in water and in saline solutions. A plausible explanation for such a protective effect is the formation of a passivating bacterial EPS coating on $CaCO_3$ grains which drastically reduced the dissolution rate of this mineral[42, 43] as it has been demonstrated that bacterial EPS played a pivotal role in controlling the dissolution rates of $CaCO_3$ minerals[42]. Similarly, the incorporation of bacterially derived organics within the $CaCO_3$ biomineral overgrowth likely adds to the increased resistance of the newly formed calcite cement against dissolution.

The incorporation of organics within this nanostructured bacterial calcite appears to be at the root cause of the mechanical strength increase and resulting consolidation effect[10]. Incorporation of organics is a general phenomenon during biomineralization and biomimetic crystallization of calcium carbonate[41, 44], which strongly affects the physicochemical properties of the resulting organic-inorganic hybrid biomaterial[44, 45]. Incorporation of small amounts of organics (a few wt%) in $CaCO_3$ biominerals significantly increases their hardness and toughness, compared to the inorganic constituent they are made of (e.g., up to three orders of magnitude increase in the work of fracture in the case of mollusk shells[46]). It has been demonstrated that the hierarchical structure of $CaCO_3$ biominerals, with a first level of organization including an assembly of nanocrystals interspersed with organics, strongly contributes to their toughness[44–46]. Such a hierarchical structure is observed here, with a first level of organization at the nanoscale (i.e., the building units are nanoparticles), and a second level of hierarchy at the microscale, represented by larger mesoestructured $CaCO_3$ particles interspersed with organics (EPS) and formed via aggregation of nanoparticles. Several toughening mechanisms have been proposed for the case of biominerals, including (micro)crack bridging[46], viscoelastic deformation of organics[47], dislocation

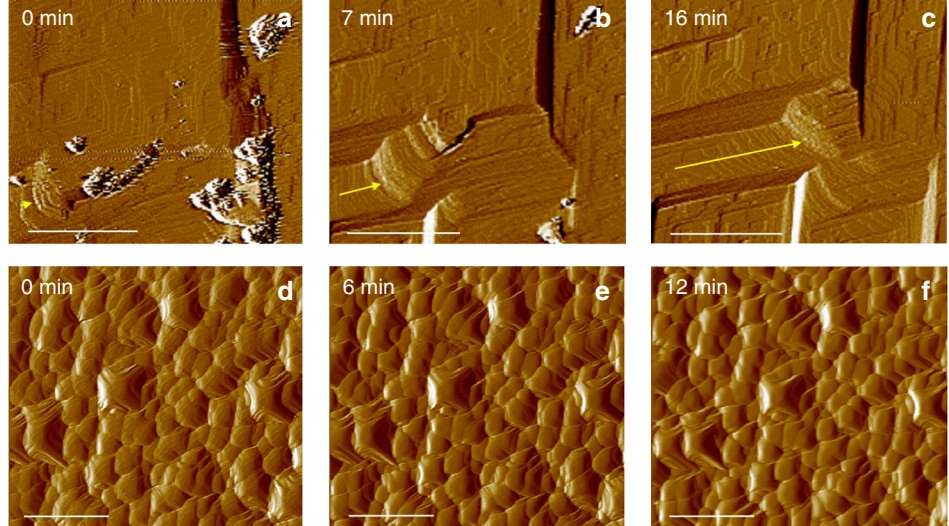

**Fig. 7** Time-resolved in situ AFM images of the dissolution of calcite. **a–c** control (untreated) and **d–f** bacterially treated (104) calcite surface exposed to flowing 1 M $MgSO_4$ solution. Time = 0 min represent the moment when the saline solution was injected into the AFM fluid cell. The *yellow arrows* in **a–c** show the retreat of the macrostep over time (due to fast calcite dissolution). Note that no dissolution features are observed in **d–f** over the time course of the dissolution experiment. *Scale bar*: 1 μm

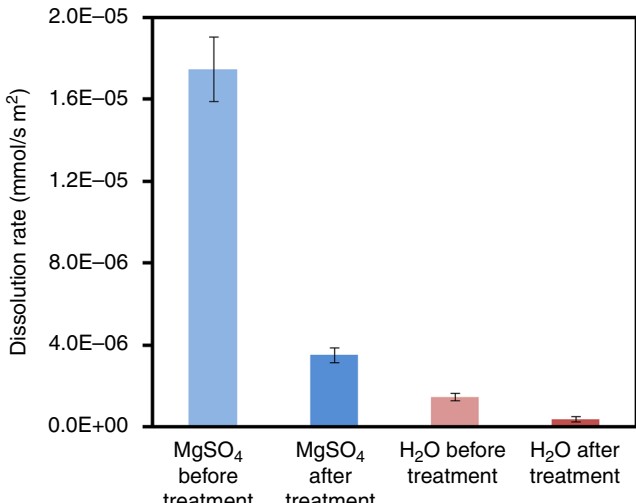

**Fig. 8** Dissolution rate of untreated and bio-treated calcite in 1 M $MgSO_4$ and $H_2O$. *Error bars* show s.d. ($\pm 1\sigma$)

pinning[44], and flaw tolerance of nanostructures during stress application[48]. An effective mechanism for stress energy dissipation is crack deflection between nanogranular units[49] and nanoparticle rotation[50] in a hybrid organic-$CaCO_3$ biomineral structure (e.g., mollusk shells). Dislocation pinning by occluded organics within calcite single crystals also contributes to a significant increase in hardness[44]. For instance, the often reported nanoindentation hardness of geologic calcite is ~ 2.5 GPa[44, 51]. In contrast, the nanoindentation hardness of biogenic and biomimetic calcite (with occluded organics) reaches values of 3.5 to 4.2 GPa[44, 51]. Remarkably, bacterially induced calcite shows a nanoindentation hardness of up to 3.92 ± 0.43 GPa[52]. These toughening effects, particularly those taking place at the nanoscale, may explain why in our study the treated weathered areas displayed higher DR values than even those of unweathered calcarenite, an effect that can not be ascribed to a porosity reduction (which is very limited, according to MIP results)[53]. Indeed, the newly formed bacterial calcite structures made up of nanounits interspersed with EPS, which strongly resembled

mesocrystals[28], resulted in a tougher cement with a high consolidation capacity (as demonstrated by peeling tests and DR results). The increased strength associated with bacterial mineralization was instrumental to prevent/limit physical weathering due to in-pore salt crystallization pressure, at least over the first 24 months after treatment application. In addition, carboxylic-rich EPS likely acts as a template for the heterogeneous nucleation of salts at a lower supersaturation, thereby reducing crystallization pressure and associated damage. This effect has been demonstrated for the case of other organic templates adsorbed onto calcite substrates which resulted in a drastic reduction of salt crystallization pressure and damage[54]. Our new self-inoculation bio-treatment may thus represent the only effective consolidation method for porous materials subjected to salt weathering, without the need of desalination, application of crystallization inhibitors/promoters, tight environmental control, or other conventional treatments with limited efficacy against salt weathering damage[26, 54].

Overall, our study demonstrates that from both a microbiological and a conservation point of view, this new self-inoculation bio-treatment is highly effective at protection and consolidation of heavily damaged calcareous stones, and results in no known side-effects. The fact that the identified carbonatogenic bacterial species in the stone indigenous microbial community are among the most common in stone artworks and monuments all over the world, may enable the presented novel bio-conservation method to have widespread applications.

## Methods

**Selection and analysis of weathered calcarenite stone.** Highly decayed porous limestone (calcarenite) blocks were selected for the testing of conservation treatments. They form part of an upright exterior wall at the main entrance of one of the two cloisters of the San Jeronimo Monastery (Granada, Spain) that has been exposed to centuries of weathering (Fig. 1a). This XVII[th] c. monument was built of a buff-colored, highly porous (~ 27%; Supplementary Fig. 6), bioclastic calcarenite stone, characterized by a limited cementation that makes it highly susceptible to weathering. Figure 1b–d shows the areas selected for in situ treatments. Small stone samples (~ 1 g sample mass) were collected for laboratory analysis from each of these sections. For comparison purposes, samples were also collected from stone blocks located ~ 1 m above these highly weathered areas, which are considered unweathered due to their limited damage. Sampling was performed prior to, and 24 months after treatment. Stone mineralogy and the relative abundance of different salts were determined by X-ray diffraction (XRD) on a PANanalytical XPert

Pro equipped with Ni filter (measurement parameters: Cu Kα radiation λ = 1.5405 Å, 45 kV, 40 mA, 4 to 70 °2θ exploration range, steps of 0.001 °2θ, and goniometer speed of 0.01 °2θ s⁻¹). Powders were deposited in zero-background Si sample holders for analysis. Mineral phases were identified by comparison with JCPDS powder spectra (Joint Committee on Powder Diffraction Standards) and quantified using the XPowder computer program[55]. Additional microstructural features of untreated and treated calcarenite samples were determined by scanning electron microscopy (Zeiss VPSEM, LEO 1430-VP and Zeiss FESEM, AURIGA) equipped with X-ray spectroscopy (EDS) microanalysis. Samples were carbon coated prior to SEM observations.

We carefully selected the treatment areas such that they all have the same stone type, the same exposure, the same level of degradation, and suffer from the same damage mechanism (salt weathering). All these prerequisites were fulfilled by the three selected calcarenite stone blocks. Indeed, our analyses corroborated that all three blocks had the same composition, porosity, drilling resistance and weight loss following the peeling tape tests (within error). They also have comparable salt composition and content, and microbiota (Results section).

**Previously studied bacterial conservation bio-treatments**. Two stone blocks showing similar extensive deterioration were subjected to well-established bacterial bio-consolidation treatments[3, 18, 19] using: (1) a 48 h-old *M. xanthus* culture as inoculant; or (2) a sterile M-3P nutritive solution (1% Bacto Casitone, 1% Ca (CH₃COO)₂.4H₂O, 0.2% K₂CO₃.1/2 H₂O in a 10 mM phosphate buffer, pH 8)[22] without the addition of any microorganism. Controls consisted of areas of the weathered stone blocks left untreated.

In the *M. xanthus* treatment, the stone was treated on the first day with the culture of *M. xanthus* and then for the remaining 5 days of treatment, only the sterile M-3P nutritive solution was applied. In the sterile M-3P treatment, only the nutritive solution was applied to the stone during the 6 days of the treatment. In both cases, the solution was sprayed onto the stone twice each day to avoid desiccation of the stone. To maintain the stone adequately damp and protected from the direct effect of sunlight, the treated areas were covered with an aluminum/plastic foil (not in direct contact with the stone surface) during the treatment and up to 3 days after treatment completion until the solutions evaporated completely. The treatment application is described in further detail in Jroundi et al.[18].

In the case of the first type of conventional treatment using a single exogenous bacterial strain, it could be argued that a *Bacillus* sp. could be a better choice than *M. xanthus* as a reference for comparison with the self-inoculation treatment studied here because *Bacillus* sp. were present in the stone carbonatogenic community (Results section) and they are among the most commonly used exogenous bacteria for microbially induced carbonate precipitation in stone[12]. However, past treatments using *Bacillus cereus* (the well-known Calcite Bioconcept treatment)[8] showed two main drawbacks, which prompted us to propose the use of *M. xanthus* for limestone conservation via microbially induced carbonate precipitation, as detailed in Rodriguez-Navarro et al.[10]. The first drawback is the poor penetration of the bacterial CaCO₃ coating (only a few micrometers), which typically led to a limited consolidation (the biocalcite layer was only 3–5 μm thick)[8]. The second, and most important drawback, is the massive formation of bacterial EPS covering and blocking the stone porous system. Although it reportedly reduced water absorption[8], overall, this is highly detrimental because it may prevent water vapor transport. Biofilms produced by *Bacillus* sp. have been shown to be hydrophobic and to limit gas permeability[56]. These effects can lead to enhanced stone deterioration, especially in the case of salt weathering, as shown by May et al.[57]. Successful treatments of porous carbonate stones using *Bacillus* spp., including *Bacillus pasteurii* (now named *Sporosarcina pasteurii*), *Bacillus subtilis*, and *Bacillus sphaericus* have been reported[12]. However, in all these cases the nutritive medium included urea (and a calcium source, typically CaCl₂) and carbonatogenesis was induced via urea-hydrolysis catalyzed by bacterial urease. One problem associated with this biomineralization route is the massive production of (toxic) ammonia, and the possible discoloration of the treated stone, as well as the potential for the activation of other deleterious microbial processes such as bacterial nitrate production[58]. Our nutritive solution does not include urea, and does not rely on urease activity for successful carbonatogenesis. Moreover, our nutritive solution is specifically designed to activate the carbonatogenic bacteria present in stones. This is not the case of urea-based nutritive media. We previously tested the isolation of indigenous *Bacillus* spp. present in decayed calcarenite stone in monuments of Granada and their application as single bacteria inoculum for the treatment of calcarenite stones. In our paper Jroundi et al.[59], we showed that three strains of stone-isolated *Bacillus* sp., which displayed a strong carbonatogenic capacity when cultured in our M-3 medium, were each effective at the strengthening of calcarenite stone blocks (laboratory tests). Sonication tests showed a weight loss reduction of up to 54% after *Bacillus* sp. treatment, compared with non-treated controls. These strengthening results are comparable to those reported by De Muynck et al.[60], who used *B. sphaericus* cultured in urea-containing nutritive medium, and observed a reduction of 63% weigh loss (using a similar sonication test) following treatment of porous limestone. In contrast, our treatment using *M. xanthus* as a foreign carbonatogenic bacterium and M-3P as a nutritive solution led to a weight loss reduction upon treatment of calcarenite stone of up to 67–72% (results of sonication tests)[21, 61]. These results confirm that *M. xanthus* can achieve a consolidation efficacy as good or better than other bio-consolidation treatments

using *Bacillus* sp. foreign or indigenous, without the drawbacks pointed out above. Note also that *M. xanthus* has shown a strong capacity for the protection and consolidation of weathered calcarenite stone once applied in situ, on different stone buildings[3]. This is due to its high carbonatogenic capacity, its ability to penetrate deep into the porous system of stone (due to its gliding motility) and its demonstrated capacity to act synergistically with the indigenous stone microbiota for an enhanced consolidation[18]. Moreover, several *Myxococcus* spp. have been found in marine environments and saline soils, and in the specific case of *M. xanthus*, it has been shown that cells can proliferate under osmotic stress[62]. Such a salt-tolerance can ensure *M. xanthus* proliferation once applied to salt weathered stones. All in all, these published results prompted us to discard the use of a *Bacillus* sp. (foreign or isolated from the treated calcarenite stones) for the treatment using a single bacterial culture and to instead use *M. xanthus*.

**The new bacterial self-inoculation bio-treatment**. The new bio-treatment consisted of the in situ application of a culture of stone-bacterial community isolated from the weathered stone and activated in the laboratory with the sterile M-3P nutritive solution (see Supplementary Fig. 7 for a detailed schematic of the treatment protocol). To obtain this culture, stone samples (~ 1 g in total) were collected from different spots of the weathered calcarenite stone surface prior to the application of any treatment. Sterile tweezers and Eppendorf vials were used during sample collection. Subsequently, stone samples were added to 5 ml of sterile nutritive solution M-3P and incubated during 48 h at 28 °C under shaking at 180 rpm. Aliquots of this activated bacterial community culture were stored in glycerol vials at −80 °C, in a 1:1 ratio (v/v) as bacterial suspensions for long-term preservation and for further molecular identification studies, and used as pre-inoculant in order to obtain a sufficient amount of a 24 h-old bacterial community culture necessary for the bio-consolidation treatment. The optical density of the 24 h-old community culture applied to the stone surface was 1.304, corresponding to ~ 8.2 ± 0.32 × 10⁸ CFU ml⁻¹. The application of this treatment was performed as described above for the other two bio-consolidation treatments. It was applied on a heavily weathered stone block, adjacent to those treated by the conventional bio-consolidation treatments described above (Fig. 1). In all the three cases, the volume of solution applied to the stone was approximately 0.125 ml cm⁻² for each application (twice a day for the 6 days of treatment), and the treatments were performed in May, a dry season when the average maximum and minimum $T$ in the area are 24 and 10 °C, respectively, and the relative humidity is typically below 50%. The warm weather was chosen to avoid the potential detrimental effects of low $T$ on the development of bacteria. The dry conditions during treatment application and subsequent in situ testing (see below) after 5, 12 and 24 months were selected to avoid any potential effects of a high humidity (i.e., associated with rainfall events) on the peeling tape and drilling resistance tests. A part of the stone block was maintained without any treatment as a control area.

**Molecular identification of the bacterial community**. Supplementary Fig. 7 schematically shows the experimental protocol for the isolation and identification of the bacterial community. An aliquot of the bacterial community culture (obtained from the bacterial suspensions kept at −80 °C) was re-freshed in sterile M-3P nutritive solution (0.5 ml bacterial population/glycerol mixture in 5 ml nutritive solution) and incubated for 24 h at 28 °C, 180 rpm. Then, serial dilutions and spreading on plates of M-3P solid medium (M-3P supplemented with 2% purified agar, Difco) were performed and the plates were incubated at 28 °C in order to obtain colonies. Whenever new colonies emerged during the course of 7 days (identified by differences in color, size or morphology), these colonies, plus some randomly selected ones, were isolated and purified by streak plating onto tryptic soy agar (TSA; Scharlau, Chemie SA, Barcelona, Spain) to ensure the isolation of pure cultures. Each isolate was then transferred to TSB liquid medium and incubated overnight at 28 °C. Cells were harvested by centrifugation (13,000 r.p.m., 8 min) into pellets and washed twice with sterile distilled water. From the obtained pellets, pure genomic DNAs were extracted as described in Martín-Platero et al.[63]. The isolates were clustered using the REP-PCR technique and the 16S rDNA of the representatives of each group were subsequently amplified as described in Jroundi et al.[18]. The sequences obtained were compared with those from the GenBank using the BLASTN computer code available at EMBL-EBI (http://www.ebi.ac.uk/). Sequences were aligned using the multiple sequence alignment program ClustalW2, also available at EMBL-EBI. All sequences were deposited in the NCBI GenBank under the accession numbers from KX301301 to KX301314, corresponding to the strains SJC6, SJC43, SJC46, SJC49, SJC50, SJC45, SJC11, SJC34, SJC22, SJC21, SJC38, SJC15, SJC16, SJC10, respectively.

**Carbonatogenic activity of the bacterial community**. To determine the carbonatogenic capacity of the members of the bacterial community isolated from the stone (the isolates obtained and identified as described above), plates of M-3P solid culture medium were inoculated with aliquots of single bacterial strains from the above-mentioned serial dilutions and incubated at 28 °C for 7 days. Controls of sterile culture medium were incubated in parallel. Petri dishes were periodically examined by optical microscopy for the presence of calcium carbonate crystals. The carbonatogenic activity was considered high when CaCO₃ precipitates were covering the whole colony and were also present in the bulk medium. The mineralogy

of the calcium carbonate precipitates formed by the bacterial community isolates was determined by XRD.

**Evaluation of the efficacy of bacterial treatments**. In situ tests at the San Jeronimo Monastery were performed before treatment and 5, 12, and 24 months after the bio-treatment application.

To quantify the change in the surface cohesion after the bio-consolidation treatments, the so-called peeling tape (or Scotch tape) test was used[64]. This test, which has been successfully utilized to test the efficacy of stone conservation treatments[3, 18], involves the measurement of the weight difference of adhesive tape pieces before and after they have been stuck to the stone surface, and calculated per unit surface area. The adhesive tape removes loose and/or poorly cemented surface carbonate grains, therefore, variations in the mass of grains removed by the tape enable quantification of the surface consolidation achieved by the treatment. Pieces of adhesive tapes ($7 \times 3.5$ cm$^2$) were stuck on the stone surface and rapidly removed. This operation was performed on the untreated surface and on the treated stone both before and after treatments and at each of sampling time after treatment. To ensure statistical significance, a minimum of 3 replicates per sampled area was performed.

Drilling resistance (DR) was measured in situ (before and after bacterial treatments) by a drilling resistance measurement system (DRMS; Sint Technology). This technology is one of the most promising for the in situ evaluation of consolidation (strengthening) performance[65]. The DRMS continuously measures the force necessary to drill a hole in a material under specific operating conditions. During testing, both rotational speed ($\omega$) and penetration rate ($\upsilon$) are maintained constant. The test conditions used in this study were: 5 mm diameter Diaber drill bit (Sint Technology) with a flat-edged diamond-tip; $\upsilon$ set to 10 mm min$^{-1}$; $\omega$ set to 300 rpm (optimal for porous calcareous stones). A calibration standard of known DR was also drilled to ensure no variation in DR associated with wear of the drill bit tip during the course of the tests. Under these conditions, the cutting depth per revolution, $\delta$ (where $\delta = 2\pi\upsilon/\omega$) is equal to 0.2 mm rev-1. A minimum of 6 drill tests were carried out on each tested area (i.e., salt weathered stone blocks shown in Fig. 1, as well as unweathered and untreated stone blocks located ~ 1 m above the weathered area). Drill holes were subsequently plugged with a lime-based conservation mortar. It should be noted that DR results show a good linear correlation with mechanical properties such as uniaxial compressive strength and biaxial flexural strength, which includes the combined effect of compressive and tensile strength[65, 66]. As expected, DR values show an exponential correlation with Mohs's hardness[65], and an exponential decay with porosity (for porous materials like limestone, gypsum plaster or lime mortars)[53].

The porosity and pore size distribution of the calcarenite stone blocks were determined by mercury intrusion porosimetry (MIP, Micromeritics Autopore III) before and after treatments. Samples (~ 1 g) were collected from the first 5 mm beneath the stone surface for analysis. Samples were dried in an oven at 80 °C for 24 h prior to analyses which were performed in duplicate.

Color changes were evaluated at each sampling time by measuring the $L^*a^*b^*$ color parameters using a Minolta Chroma Meter portable spectrophotometer equipped with Xenon lamp (Illuminant C) and diffuse reflectance geometry[3]. Total color variations were reported as $\Delta E = \Delta(L^{*2} + \Delta a^{*2} + \Delta b^{*2})^{1/2}$, where $\Delta L^*$, $\Delta a^*$ and $\Delta b^*$ are, respectively, the difference in values between the untreated and treated stone of: $L^*$ (lightness: 0 being black and 100 being diffuse white), $a^*$ (negative values indicate green whilst positive values indicate magenta), and $b^*$ (negative values indicate blue and positive values indicate yellow). To ensure statistical significance, a minimum of 3 color measurements was performed per test area. The statistical significance of color measurements (before and after treatment) was evaluated by using the $t$-test implemented in the Statgraphics Centurion XVI.II code (Statpoint Technologies, Inc., The Plains, Virginia, USA).

**Evolution of stone microbiota following treatment**. The evolution of the culturable microbiota was tracked over several months (prior to treatment and 5, 12, and 24 months after), to determine if the treatments activated any deleterious microbiota. At each sampling time, stone grains from treated and untreated areas (~ 300 mg per sample) were collected and added to 1 ml of sterile 0.9 wt% NaCl solution. Each sample was then gently mixed, and aliquots were collected for serial dilution, from which different culture media were inoculated. The total number of bacteria was assessed using TSA culture medium (Scharlau Microbiology, Spain); organic acid-producing bacteria were detected and tested using Hugh and Leifson culture medium;[67] nitrifying bacteria were tested using the culture media of Soriano and Walke[68] and Aleem and Alexander;[69] sulfur-oxidizing bacteria were tested using the culture medium of Johnson and Peck;[70] finally, the total number of fungi was tested by using Sabouraud culture medium (Scharlau Microbiology, Spain).

**Effectiveness against salt-enhanced chemical weathering**. Nanoscale in situ atomic force microscopy (AFM; Digital Instruments Nanoscope III Multimode) analysis was performed on the dissolution of calcite single crystals before and after being subjected to the bacterial community bio-treatment. Dissolution experiments were performed in a fluid cell of the AFM with a scanning frequency of 4 Hz (giving an average scan time of 1.5 scans per minute) and the area scanned was

mostly $3.5 \times 3.5$ μm or $10 \times 10$ μm. Images were collected in contact mode using a Si$_3$N$_4$ tip (Veeco Instruments, tip model NP-S20) with a spring constant of 0.12 N m$^{-1}$ under ambient temperature (20 °C). Using a syringe, deionized water (resistivity ~ 18.2 MΩcm), 1 M MgSO$_4$ or 1 M NaCl saline solutions were injected in the fluid cell at an average flow rate of 60 ml h$^{-1}$. Prior to each AFM experiment, optical quality calcite crystals (Chihuahua, Mexico) were cleaved using a razor blade to expose fresh {10$\bar{1}$4} planes and mm-sized fragments (~ $2 \times 3 \times 5$ mm) were obtained. The calcite crystals were first sterilized by tyndallization (flowing steam for 1 h at 100 °C, four times in succession at 24 h intervals). Each crystal was then added to a test tube containing 5 ml of sterile M-3P culture medium and 0.1 ml of bacterial community inoculum and incubated for 7 days at 28 °C with shaking (160 r.p.m.) using a rotary Certomat R shaker (Braun). Controls were incubated in parallel without the addition of inoculum. A minimum of three replicate samples were prepared. The treated crystals were rinsed in deionized water, dried at room $T$, and stored in Eppendorf tubes until analyses.

Macroscale dissolution experiments were performed using untreated (control) and bio-treated (treatment with the bacterial community as described above) powdered (sterilized) calcite grains (1 g, 63–125 μm in size obtained from ground Macael marble) placed in a $T$-controlled (25 °C) flow-though Teflon reactor (Supplementary Fig. 8). A Gilson Minipuls Evolution peristaltic pump supplied a constant flow of 0.147 ml min$^{-1}$ (a sufficiently high flow rate that the solution never reached saturation with respect to calcite and the dissolution kinetics remained surface, rather than diffusion controlled). The outlet solution was collected every 2 h at the start of an experiment and then daily for a 2 h period until completion of the experiment (7 days). The salt solutions used in these experiments were prepared at an ionic strength of 1 from anhydrous salts (Panreac Quimica Sau) and deionized water in equilibrium with air. The Ca$^{2+}$ content of the output solutions was measured using inductively coupled plasma-optical emission spectroscopy (ICP-OES, Varian Vista proaxial). Dissolution rates were normalized to calcite powder surface area (BET method), which was measured before and after dissolution experiments using N$_2$ adsorption (Micromeritics TRISTAR 3000). $T$-tests were implemented to evaluate the statistical significance of the changes in dissolution rate taking place after bio-treatment.

**Analysis of bacterial community CaCO$_3$ mineralization**. 5 ml of sterile M-3P culture medium and 0.1 ml of bacterial inoculum were incubated in test tubes for up to 48 h at 28 °C. Precipitates were collected at predetermined time intervals, rinsed in deionized water, dried at room $T$, and stored in plastic vials until analysis. The bacterial-mineral assemblages were studied using a transmission electron microscope (TEM; Philips CM20), operated at 200 kV with a 40 μm objective aperture. Prior to analysis, bacterial precipitates were dispersed in ethanol and deposited on carbon/Formvar® film coated copper grids. Selected area electron diffraction (SAED) patterns were collected using a 10 μm aperture, which allowed collection of diffraction data from a circular area ~ 0.2 μm in diameter. The bacterial-mineral assemblages were also studied using a SEM (Zeiss Gemini Ultra-55 with field emission gun), equipped with transmission (STEM) bright/dark field detector compatible with standard TEM grids and working at 30 kV.

Selected calcite single crystals bio-treated with the bacterial community corresponding to the same batch used for AFM in situ dissolution analyses (see above) were collected (after 17 h, 24 h, 72 h, and 7 days), C coated and studied using FESEM (Zeiss Gemini Ultra-55 and AURIGA).

**Data availability**. The data that support the findings of this study are available from the corresponding author (C.R.-N.) upon reasonable request.

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

## Acknowledgements

This work was supported by the Spanish Government (Grants MAT2012-37584, CGL2012-35992 and CGL2015-70642-R), the Junta de Andalucía through Proyecto de excelencia RNM-3493 and Project P11-RNM-7550, the Research Groups BIO 103 and RNM-179, and the University of Granada (Unidad Científica de Excelencia UCE-PP2016-05). Additional funds were provided by the Molecular Foundry (Lawrence Berkeley National Laboratory, LBNL, University of California, Berkeley, CA) for a research stay of M.S. (project #1451; User Agreement No. NPUSR009206). We thank the personnel of the Centro de Instrumentación Científica (CIC; University of Granada) and the Molecular Foundry (LBNL, University of California Berkeley) for analytical assistance.

## Author contributions

M.T.G.-M. and C.R.-N. conceived the concept and designed the experiments. All authors contributed to perform the experiments and the analysis of results. F.J., M.S. and C.R.-N. wrote the paper with contributions from all authors.

## Additional information

**Competing interests:** The authors declare no competing financial interests.

