## [Peer Review file · Nature Communications]

Comments on “Protection and consolidation of stone heritage...” F. Jroundi by *et al.*

MS # 111791_0_art_file_2010206_vflnn5

This is an excellent piece of work – important, thorough, and generally well presented. My only comments are in regard to the clarity of the explanation.

1. If I understand this correctly, the authors identified the range of species in the sample, but they used the entire colony in the inoculation. That is, they did not isolate and employ specific carbonatogenic species. If so, why not separate them? Is it to preserve some symbiotic relationship among the species, or is it too difficult to separate them? The wording seems ambiguous throughout the paper. For example, in line 168 it says “the isolated and activated bacterial community was inoculated”. Does “isolated” mean that a specific group of species was chosen?
2. If the entire colony was employed, why was it so much more effective to culture them externally and reintroduce them, rather than to inject the M-3P medium into the stone?
3. In line 236 it is reported that the treated surface has a drilling resistance 4 times as high as the untreated and unweathered stone. Is that perhaps too high? Many conservators object to creating a strong discontinuity in properties, for fear that the treated layer will separate.
4. In line 293 it is reported that the population of acid-producing bacteria decreased. Is it known why this occurred? Is it to be expected in general or is it just good luck in this case?
5. There is no discussion of the mechanism of growth of the nanoparticulate calcite, although that might be explained in the cited literature. Presumably, there must be a contiguous pore network leading to the surface of the bacterium to allow nutrients to enter and the excreted carbonates to exit. Do the small pores constitute a risk with respect to salt damage, since they favor the generation of higher crystallization pressure? Or do the pores eventually become filled with organics?
6. The duration of the tests described in Figure 8 should be cited near line 335, not just in the later description of experimental procedure. Figure 7 shows a duration of 12 minutes, which would not necessarily indicate a useful increase in protection, but the week-long tests used to generate Figure 8 are more reassuring.
7. Line 428 refers to the role of hierarchical structure in toughening of natural composites, such as abalone shells. Is there any indication of such structure in the present case?

Reviewer #2 (Remarks to the Author):

This manuscript describes a new methodology to consolidate stone using a bacterial community. Previously either a single culture or a sterile medium to activate the carbonatogenic bacteria have been used. Further, the authors proved that their community can be applied to salt weathered stone.

Major remarks

- I would not be able to apply this new methodology as some information is missing. Do you mix the cultures before application or do you apply each activated culture sequentially? How much volume is applied per square centimeter? What is the cellular density? Would you apply all the cultures grown in your medium even if not all of them show high carbonatogenic activity? What do you consider the threshold between high and low carbonatogenic activity? Do you think that the temperature at the time of application can affect the treatment? Any other environmental parameter to take into account? (Lines 367-371 and 508-523).
- Lines 205-206. Three areas were chosen for the treatments with *M. xanthus*, the M-3P nutritive solution, and the self-inoculation. The authors should better explain how they proved the areas selected for the three treatments had the same/similar state of conservation. This is an important claim to make. Indeed, if the stone treated with the bacterial community was in a better state of conservation, this could partially explain the superior results of the proposed approach.
- The authors identified many *Bacillus* spp. on stone of the San Jeronimo Monastery. No *Myxococcus* sp. was found. In my opinion a better comparison with the self-inoculation with the indigenous carbonatogenic bacterial community would have been the choice of a *Bacillus* sp. (e.g. one retrieved from the same monument or one of the many reported in the literature for consolidation). Please explain the choice of a *Myxococcus* sp.
- Line 68. Please provide a reference for this statement.

Minor remarks

Abstract.

Line 12. Stone monuments are artworks. Please change the sentence to "... stone monuments and other artworks".

Lines 12-13. Unfortunately, it is not possible to stop the ageing of stone but only to slow down the process. Please change "To halt these deleterious processes and mitigate their effects" to "To mitigate weathering processes".

Introduction.

Lines 33 -34. Unfortunately, some of these processes are neither slow nor linear.

Lines 38-44. You can add that even synthetic polymers used for protecting or consolidating stone can be attacked by microorganisms. A couple of fine reviews on this topic are available.

Lines 63-65. It depends on the bacterial load applied.

Line 80. How the fact that the medium is patented can affect its use by other researchers?

Results.

Lines 278-279. The number of bacteria is not necessarily related to bacterial activity. Cells can be numerous but in a quiescent state.

Line 286. I would change "population" to "community".

Fig. 8. Numbers should be subscripts.

Reviewer #3 (Remarks to the Author):

Protection and consolidation of stone heritage by self-inoculation with indigenous carbonatogenic bacterial communities

Fadwa Jroundi¹, Mara Schiro,² Encarnación Ruiz-Agudo,² Kerstin Elert,² Inés Martín-Sánchez,¹ María Teresa González-Muñoz¹ and Carlos Rodríguez-Navarro^{2*}

Nature communications – Review

This is an interesting paper building further upon the earlier research of this research group, where consolidation of limestone was achieved by (1) application of exogenous single bacteria cultures, or (2) by application of nutrients to activate the carbonatogenic bacteria in the microbial community of the stone itself. Now, it is suggested (3) to extract the indigenous bacterial community from the carbonate stones that have to be consolidated, activate the carbonatogenic bacteria, and then apply this activated culture again to the same stone. The aim of the authors is to realize a better consolidation of salt weathered stones.

The authors in this paper compare the 3 previously mentioned techniques and show that the 3rd one leads to a better stone consolidation. They use a combination of state-of-the-art methods to support their findings. The paper supplies new findings and is well written. Therefore, in principle, I support publication.

Nevertheless, I have one major comment regarding the comparison between the 3 techniques. The bacteria isolated from the stone in the new technique (3), are mainly *Bacillus* species, which are known to have often a high carbonatogenic activity and to be halotolerant. They have been used before by many researchers for consolidation of stones and soils. Nevertheless, in technique (1) *M. xanthus* is applied, and it is shown that the effect of technique (1) is less effective than (3). According to me, the comparison is therefore not valid: a comparison should have been made with technique (1) using application of *Bacillus* species coming from another (exogenous) source. Only when a comparison is made with application of similar bacteria, that have proven their consolidation potential, but exogenous from the actual stones to be consolidated, it would be proven that technique (3) with indigenous bacteria is better than (1). So the authors should either do these additional experiments, or change their conclusions and state that good consolidation was obtained by technique (3) without claiming that the technique itself is necessarily better than (1). Of course the first option (doing the additional tests) would be preferable, but may not be possible at this stage.

Some other comments are:

Lines 63-65: It is mentioned that exogenous bacteria are likely to be at a competitive disadvantage. But even if bacteria are isolated from the stone itself and one strain is stimulated and applied again, will this stimulated strain not also be pushed back again by the other bacteria present to move towards the original equilibrium? Would there be really a difference with exogenous bacteria (obtained from another source but in reasonably similar conditions)?

Lines 101-103: the differences in DR between weathered and unweathered stones are really small (in comparison to the large standard deviation). Are these differences significant? The same remark can be made for Fig 5.

Fig 6. Do the XRD peaks point at more quartz after treatment? Why would this be?

Line 291. The "f" after UFC/g can be omitted?

Line 355. If there is no pore plugging at all (suppl fig 6 even points at pore coarsening?), is it not expected that the salts will easily penetrate to the unprotected zone underneath the outer few mm of the stone?

Line 356. How was it proven that CaCO₃ was "abundant"?

Complete discussion part: some paragraphs could probably be removed since there is no additional information that is not already stated in the results part (e.g. check lines 397-401; 407-409; 361-

371)

Line 561-566: had all stones similar humidity at the moment of the tape test? The humidity could affect the adhesion of the tape.

Reviewer #1:

This is an excellent piece of work – important, thorough, and generally well presented.
We thank this referee for his very positive overall impression of our work.

My only comments are in regard to the clarity of the explanation.

1. If I understand this correctly, the authors identified the range of species in the sample, but they used the entire colony in the inoculation. That is, they did not isolate and employ specific carbonatogenic species. If so, why not separate them? Is it to preserve some symbiotic relationship among the species, or is it too difficult to separate them? The wording seems ambiguous throughout the paper. For example, in line 168 it says "the isolated and activated bacterial community was inoculated". Does "isolated" mean that a specific group of species was chosen?

The referee is right. Our wording at different parts of the manuscript was indeed ambiguous. We used the entire community of carbonatogenic bacteria isolated from the building stones and activated (i.e., cultured) in the laboratory using the M-3P nutritive solution. We now explicitly state so in the Methods section and throughout the Ms. (see also the new Supplementary Fig. 7).

Although we isolated in the laboratory each individual bacteria species present in the stone indigenous community to identify its carbonatogenic capacity (as we stated in the Methods), we did not inoculate them separately during the *in situ* treatment (applied to the building stones). As pointed out by the referee, we applied the whole community of carbonatogenic bacteria to preserve the symbiotic relationship of these bacteria (which were already present in the building). The application of each individual isolated bacterial species on a different stone block would have been impractical (we identified more than 20 different carbonatogenic bacterial species), and would miss the objective of this study, which was to test the ability of a self-inoculation treatment with the indigenous carbonatogenic bacterial community to consolidate and protect the stones. If each individual bacterial species would have been applied separately, the treatment would have fallen under the first category (type 1 treatment): i.e., application of an individual (in this case indigenous) bacteria.

2. If the entire colony was employed, why was it so much more effective to culture them externally and reintroduce them, rather than to inject the M-3P medium into the stone?

The effectiveness of the application of the entire carbonatogenic community was related to the fact that the activated bacterial inoculum was at the end of the exponential growth phase, meaning that the bacterial cells were in their optimal metabolic activity. Thus, right from the beginning of the treatment application, a large number of carbonatogenic bacterial cells were able to contribute (via their metabolic activity) to induce calcium carbonate precipitation. Note that the effectiveness of any bacterial conservation treatment via calcium carbonate precipitation is directly related to the bacterial cell load (Ref. #12: De Muynck et al., 2010 Ecol. Eng. 36, 118–136). This point has been clarified in the revised version where we now state: "*These results show that the laboratory activation of the isolated carbonatogenic bacterial community cultured for 24h in the M-3P nutritive solution (i.e., at the final stage of the exponential growth) prior to the in situ application (i.e., "self-inoculation") of the treatment solution was critical to the effectiveness of this treatment. Right from the beginning of the treatment application, a large number of carbonatogenic bacteria were able to contribute (via their metabolic activity) to induce calcium carbonate precipitation. Note that the effectiveness of any bacterial conservation treatment via calcium carbonate precipitation is directly related to the carbonatogenic bacterial cell load.*"¹²

Note that in the previous version of the Ms. we stated that the bacterial community was cultured for 48h (i.e., after the end of the exponential growth stage). This was a mistake: they

were cultured for 24h (i.e., up to the point where a deceleration begins to be evident during the exponential growth stage). This error has been corrected in the revised version.

3. In line 236 it is reported that the treated surface has a drilling resistance 4 times as high as the untreated and unweathered stone. Is that perhaps too high? Many conservators object to creating a strong discontinuity in properties, for fear that the treated layer will separate. Such an increase in drilling resistance is indeed impressive. However, our long term monitoring of the treated stone blocks (two years) shows that such a strengthening was not detrimental. Indeed, such a remarkable strengthening was at the root of the high effectiveness of this novel bacterial conservation treatment. We now explicitly address this point by stating in the text: *"While such an increase in drilling resistance might appear excessive and could potentially lead to detrimental differences in mechanical behavior between the consolidated part and the unweathered stone, the fact that two years after the treatment application no scaling or surface material loss was observed (see peeling tape test results, above), corroborates that the strengthening achieved was pivotal for the successful consolidation of the stone."*

4. In line 293 it is reported that the population of acid-producing bacteria decreased. Is it known why this occurred? Is it to be expected in general or is it just good luck in this case? Our previous studies have shown that this is a general case when carbonatogenic bacteria are activated following the application of M-3P solution (e.g., Ref # 18: Jroundi et al., 2010 Microbial Ecol. 60, 39–54). Our explanation is that under a high proliferation of carbonatogenic bacteria, acid producing bacteria are at a competitive disadvantage. Moreover, acid producing bacteria typically use carbohydrates as energy source, and our nutritive medium lacks carbohydrates. So, the overall effect is that their cell number is drastically reduced, as observed here. To clarify this point we now indicate in the text that: *"The reduction in the population of acid-producing bacteria following treatment is likely due to the fact that they are at a competitive disadvantage compared with activated carbonatogenic bacteria.¹⁸ This situation is favored by the fact that acid-producing bacteria typically use carbohydrates as an energy source, and our nutritive M-3P medium lacks carbohydrates."*

5. There is no discussion of the mechanism of growth of the nanoparticulate calcite, although that might be explained in the cited literature. Presumably, there must be a contiguous pore network leading to the surface of the bacterium to allow nutrients to enter and the excreted carbonates to exit. Do the small pores constitute a risk with respect to salt damage, since they favor the generation of higher crystallization pressure? Or do the pores eventually become filled with organics?

The growth of bacterial CaCO_3 has indeed been explained in the cited literature (e.g., Ref. #12). Nutrient transport and the formation of precipitates surrounding the bacterial cells (shown by our SEM and TEM observations) must take place until a point when the cell is entombed (as we showed in our paper Ref. #10: Rodriguez-Navarro et al. 2003 Appl. Environm. Microbiol. 69, 2182–2193). There is no evidence showing that a new class of small pores is formed during the bacterial mineralization process. As suggested by the referee, pores in the newly formed carbonate structures enabling nutrient/ions transfer between the bacterial cells and the surrounding medium were likely sealed during the calcification/entombment process (indeed our image of an entombed cell in Fig. 4c shows no evidence for porosity in the surrounding carbonate). Our mercury intrusion porosimetry (MIP) results (Supplementary Fig. 6) show that the reduction in porosity from ~27 to ~25 % mainly took place by the partial filling of the largest pores (i.e., those with radius ~10 μm), leading to a slight reduction of the main pore size to a value of ~5 μm . Such new pore sizes would not negatively affect the stone susceptibility towards salt damage (see Ref. # 25:

Rodriguez-Navarro and Doehne, 1999 Earth Surf. Processes Landforms 24, 191-209). Moreover, our MIP results show a partial filling of pores $< 1 \mu\text{m}$ (those that most negatively might affect salt weathering), whose amount is reduced nearly by half (see Supplementary Fig. 6). In any case, such porosity variations do not affect the stone susceptibility towards salt damage, as demonstrated by the *in situ* testing of the stone physical-mechanical properties (DR and peeling tape tests) over a period of 24 months. We clarify this point in the revised version of the Ms. where we state: "*Formation of precipitates surrounding the bacterial cells (see SEM and TEM results) takes place until cell entombment.¹² Transient pores must be present in the newly-formed carbonate structures surrounding the bacterial cells to enable transport of nutrients/ions and metabolic by-products (e.g., EPS, NH_3 and CO_2). However, the eventual sealing of such small pores during entombment would stop the biomineralization process due to cell death.¹² This is consistent with our STEM observations (Fig. 4c) showing entombed bacterial cells surrounded by pore-free calcite/EPS and our MIP results (Supplementary Fig. 6) showing a reduction, rather than an increase, in the smaller pores $< 1 \mu\text{m}$.*"

6. The duration of the tests described in Figure 8 should be cited near line 335, not just in the later description of experimental procedure. Figure 7 shows a duration of 12 minutes, which would not necessarily indicate a useful increase in protection, but the week-long tests used to generate Figure 8 are more reassuring.

In the main text we now indicate the duration of the tests (one week) described in Figure 8. We believe Figure 7 shows a useful increase in protection because at the scale at which the experiment was performed (notice the $1 \mu\text{m}$ scale bar), there is a dramatic visual difference in the dissolution rates between the untreated and treated samples even within the short duration of the experiment.

7. Line 428 refers to the role of hierarchical structure in toughening of natural composites, such as abalone shells. Is there any indication of such structure in the present case?

Yes. We failed to explicitly indicate so. Indeed our TEM and SEM analyses clearly show that the bacterial biominerals have a first level of organization at the nanoscale (i.e., the building units are nanoparticles), and a second level of hierarchy at the microscale, represented by larger "mesostructured" CaCO_3 particles interspersed with organics (EPS) and formed via aggregation of nanoparticles. We state so in the revised version of the Ms.

Reviewer #2 (Remarks to the Author):

This manuscript describes a new methodology to consolidate stone using a bacterial community. Previously either a single culture or a sterile medium to activate the carbonatogenic bacteria have been used. Further, the authors proved that their community can be applied to salt weathered stone.

Major remarks

- I would not be able to apply this new methodology as some information is missing. Do you mix the cultures before application or do you apply each activated culture sequentially? How much volume is applied per square centimeter? What is the cellular density? Would you apply all the cultures grown in your medium even if not all of them show high carbonatogenic activity? What do you consider the threshold between high and low carbonatogenic activity? Do you think that the temperature at the time of application can affect the treatment? Any other environmental parameter to take into account? (Lines 367-371 and 508-523).

We thank this referee for calling our attention to such an important omission. There were indeed several details of the methodology that we failed to explain in sufficient detail. In the revised version (Methods section) we present a substantially more detailed description of the

methodology that addresses every single question that this referee (or any other person applying our new methodology) might have. To further clarify the experimental protocol for the isolation, culture and application of our new method, an additional figure is now included in the Supplementary Figures (Supplementary Fig. 7), which schematically shows the process for the self-inoculation treatment and for the identification of the bacterial community.

- Lines 205-206. Three areas were chosen for the treatments with *M. xanthus*, the M-3P nutritive solution, and the self-inoculation. The authors should better explain how they proved the areas selected for the three treatments had the same/similar state of conservation. This is an important claim to make. Indeed, if the stone treated with the bacterial community was in a better state of conservation, this could partially explain the superior results of the proposed approach.

We agree: to have areas of equal level of degradation, exposure and stone characteristics is critical for any meaningful comparison among the different treatments. Indeed, we carefully selected the treatment areas in order to have: a) same stone type; b) same exposure; c) same level of degradation, and d) same damage mechanism (salt weathering). All these prerequisites were fulfilled by the three selected calcarenite stone blocks. Indeed, our analysis corroborated that all three blocks have the same composition, porosity, DR and weight loss following the peeling tape test (within error). They also have comparable salt composition and content, and microbiota. We are confident that in this respect our results are sound. In the revised version of the Ms we now emphasize this point when describing the selection of tested areas (Methods section).

- The authors identified many *Bacillus* spp. on stone of the San Jeronimo Monastery. No *Myxococcus* sp. was found. In my opinion a better comparison with the self-inoculation with the indigenous carbonatogenic bacterial community would have been the choice of a *Bacillus* sp. (e.g. one retrieved from the same monument or one of the many reported in the literature for consolidation). Please explain the choice of a *Myxococcus* sp.

First, we want to apologize for the fact that we did not explain our choice of a *Myxococcus* sp. for the type 1 treatment (inoculation of a foreign single bacterium). That has created some confusion which is also reflected by one of the comments by Reviewer #3 and the comment by the Editor. Indeed, the calcarenite stones at the test site (as well as the stones of several historical buildings in Granada, built using the same type of calcarenite) included several *Bacillus* spp. However, we selected *M. xanthus* as the foreign bacteria for type 1 treatment (i.e., application of a foreign single bacteria inoculum) instead of a *Bacillus* sp. (either foreign or present in the stone microbiota), based on the following reasons (which are now stated in the revised Ms, Methods section):

a) Past treatments using *Bacillus cereus* (the well-known Calcite Bioconcept treatment: see Ref. #8: Le Metayer-Levrel et al. Sediment. Geol. 126, 25–34, 1999) showed two main drawbacks, which actually prompted us to propose the use of *M. xanthus* for limestone conservation via microbially induced carbonate precipitation, as we detailed in our paper Ref. #10: Rodriguez-Navarro et al., 2003. Appl. Environm. Microbiol. 69, 2182-2193. The first drawback of such a *Bacillus* sp. treatment is the poor penetration of the bacterial CaCO₃ coating (only a few micrometers), which typically led to a limited consolidation (the biocalcite layer was only 3-5 µm thick) (see Ref. #8). The second, and most important drawback is the massive formation of bacterial EPS covering and blocking the porous system. Although it reportedly reduced water absorption (Ref. #8), overall, this is highly detrimental because it may prevent water vapor transport (i.e., biofilms produced by *Bacillus* sp. have been shown to be hydrophobic and to limit gas permeability: see new Ref. #56: Epstein et al., 2011. Bacterial biofilm shows persistent resistance to liquid wetting and gas penetration. Proc. Natl. Acad. Sci. 108, 995-1000), leading to enhanced deterioration, especially in the case of salt weathering. Indeed, May et al. (new Ref. #57) have experimentally shown that

massive microbial biofilm blocking stone pores can lead to enhanced material loss due to salt weathering in limestone and dolostone.

b) There are several publications that report successful treatment of porous carbonate stones using *Bacillus* spp., including *Bacillus pasteurii* (now named *Sporosarcina pasteurii*), *Bacillus subtilis*, and *Bacillus sphaericus* (see review by De Muynck et al. *Ecolog. Eng.*, 36, 2010, 118-136; Ref. #12). However, in all these reported applications, the nutritive medium included urea (and a calcium source, typically CaCl_2) and carbonatogenesis was induced by urea-hydrolysis catalyzed by bacterial urease. One problem associated with this biomineralization route is the massive production of (toxic) ammonia, and the possible discoloration of the treated stone, as well as the potential for the activation of other deleterious microbial processes such as bacterial nitrate production (see for instance the review by Dhami et al., *Frontiers Microbiol.* 5, 304, 2014 -new Ref. #58-; or the patent by De Muynck, High performance biodeposition for strengthening of materials, WO2013/120847A1, who stated "*this process results in high amounts of ammonium which is not only undesirable from an environmental point of view but can also result in nitric acid production (upon nitrification), and hence, damage to a building material such as stone*"). Our nutritive solution does not include urea, and does not rely on urease activity for successful carbonatogenesis. It relies on amino acids degradation via bacterial ammonification, which results in very limited production of ammonia (sufficient for alkalization and carbonate precipitation, but not enough to pose any side effects). Moreover, our nutritive medium is specifically designed to activate the carbonatogenic bacteria present in stones. This is not the case of urea-based nutritive media.

c) We actually tested the isolation of indigenous *Bacillus* spp. present in decayed calcarenite stone and their application as single bacteria inoculum for the treatment of calcarenite stones. In our paper Jroundi et al. (new Ref. #59: Stone-isolated carbonatogenic bacteria as inoculants in bioconsolidation treatments for historical limestone. *Sci. Total Environm.* 425, 89-98, 2012) we showed that three individual strains of stone-isolated *Bacillus* sp., which displayed a strong carbonatogenic capacity when cultured in our M-3 medium, were effective for the strengthening of calcarenite stone blocks (laboratory tests), as demonstrated by sonication tests. These tests showed a weight loss reduction of up to 54% upon *Bacillus* sp. treatment, compared with non-treated controls. These strengthening results are nearly comparable to those reported by De Muynck et al. (New Ref. #60: *Ecol. Eng.* 36, 99-111, 2010) who used *B. sphaericus* cultured in urea-containing nutritive medium, and observed a reduction of 63% weight loss (using a similar sonication test) following treatment of porous limestone. In contrast, our treatment using *M. xanthus* as a foreign carbonatogenic bacteria and M-3P as a nutritive solution led to a weight loss reduction of up to 67-72% upon treatment of calcarenite stone (Ref. #21: Jimenez-Lopez et al., 2007, *Chemosphere* 68, 1929-1936; New Ref. #61: Jimenez-Lopez et al., 2008, *Int. Biodeterioration Biodegradation* 62, 352-363). These results confirm that *M. xanthus* can achieve a consolidation efficacy as good as or better than other bioconsolidation treatments using *Bacillus* spp. (foreign or indigenous) and without the drawbacks pointed out above.

d) Note also that *M. xanthus* has shown a strong capacity for the protection and consolidation of weathered calcarenite stone once applied *in situ*, on different stone buildings (see Ref. #3). This is due to its high carbonatogenic capacity, its ability to penetrate deep into the porous system of stone (due to its gliding motility) and its demonstrated capacity to act synergistically with the indigenous stone macrobiota for an enhanced consolidation (see Ref. #19). Moreover, several *Myxococcus* sp. have been found in marine environments and saline soils, and in the specific case of *M. xanthus*, it has been shown that bacterial cells can proliferate under osmotic stress (new Ref. #62). Such a salt-tolerance can ensure their proliferation once applied to salt weathered stones.

All in all, these published results prompted us to discard the use of a *Bacillus* sp. (foreign or

isolated from the treated calcarenite stones) for the type 1 treatment and to instead use *M. xanthus*.

- Line 68. Please provide a reference for this statement.

We have changed this sentence, and now we state: *"Most importantly, we show here that both strategies have limited efficacy when applied to heavily salt weathered stone."*

Minor remarks

Abstract.

Line 12. Stone monuments are artworks. Please change the sentence to "... stone monuments and other artworks".

We have changed this sentence to read *"Stone monuments and other artworks"*

Lines 12-13. Unfortunately, it is not possible to stop the ageing of stone but only to slow down the process. Please change "To halt these deleterious processes and mitigate their effects" to "To mitigate weathering processes".

Done

Introduction.

Lines 33 -34. Unfortunately, some of these processes are neither slow not linear.

We agree. We have removed the word "slowly"

Lines 38-44. You can add that even synthetic polymers used for protecting or consolidating stone can be attacked by microorganisms. A couple of fine reviews on this topic are available.

Thanks for the suggestion. We now state: *"Moreover, some of these conventional treatments are not environmentally friendly because they release toxic chemicals, and the treatments themselves can undergo degradation (e.g., biodeterioration of organic polymers).²"*

Lines 63-65. It depends on the bacterial load applied.

We agree. We now state so.

Line 80. How the fact that the medium is patented can affect its use by other researchers?

The patent is currently only extended to Spain, so there is no international restriction for the use of the culture medium. Moreover, KBYO Biological S.L. (a Spanish company: <http://kbyobiological.com/>) is commercially producing M-3P medium, so it could be purchased by any interested conservator/researcher worldwide.

Results.

Lines 278-279. The number of bacteria is not necessarily related to bacterial activity. Cells can be numerous but in a quiescent state.

We agree. We now state *" To help link the amount of bacteria in the stone with the observed effectiveness of the treatments, the total bacterial load was monitored before and after treatment."*

Line 286. I would change "population" to "community".

Done

Fig. 8. Numbers should be subscripts.

Done

Reviewer #3 (Remarks to the Author):

Protection and consolidation of stone heritage by self-inoculation with indigenous carbonatogenic bacterial communities

Fadwa Jroundi¹, Mara Schiro,² Encarnación Ruiz-Agudo,² Kerstin Elert,² Inés Martín-Sánchez,¹ María Teresa Gonzalez-Muñoz¹ and Carlos Rodriguez-Navarro^{2*}

Nature communications – Review

This is an interesting paper building further upon the earlier research of this research group, where consolidation of limestone was achieved by (1) application of exogenous single bacteria cultures, or (2) by application of nutrients to activate the carbonatogenic bacteria in the microbial community of the stone itself. Now, it is suggested (3) to extract the indigenous bacterial community from the carbonate stones that have to be consolidated, activate the carbonatogenic bacteria, and then apply this activated culture again to the same stone. The aim of the authors is to realize a better consolidation of salt weathered stones.

The authors in this paper compare the 3 previously mentioned techniques and show that the 3rd one leads to a better stone consolidation. They use a combination of state-of-the-art methods to support their findings. The paper supplies new findings and is well written. Therefore, in principle, I support publication.

Nevertheless, I have one major comment regarding the comparison between the 3 techniques. The bacteria isolated from the stone in the new technique (3), are mainly *Bacillus* species, which are known to have often a high carbonatogenic activity and to be halotolerant. They have been used before by many researchers for consolidation of stones and soils. Nevertheless, in technique (1) *M. xanthus* is applied, and it is shown that the effect of technique (1) is less effective than (3). According to me, the comparison is therefore not valid: a comparison should have been made with technique (1) using application of *Bacillus* species coming from another (exogenous) source. Only when a comparison is made with application of similar bacteria, that have proven their consolidation potential, but exogenous from the actual stones to be consolidated, it would be proven that technique (3) with indigenous bacteria is better than (1). So the authors should either do these additional experiments, or change their conclusions and state that good consolidation was obtained by technique (3) without claiming that the technique itself is necessarily better than (1). Of course the first option (doing the additional tests) would be preferable, but may not be possible at this stage.

Please see our answer to referee #2 where we justify the use of *M. xanthus* for type 1 treatment. We agree that under different circumstances (e.g., different stone support, environment, and salt weathering processes) a treatment using another halotolerant carbonatogenic bacterium (for instance, a *Bacillus* sp.) might also be effective. But note that in our final conclusion paragraph we do not state that this treatment has to be necessarily better than (1) in all circumstances. What we state is "Overall, our study demonstrates that from both a microbiological and a conservation point of view, this new self-inoculation bio-treatment is highly effective at protection and consolidation of heavily damaged calcareous stones, and results in no known side-effects." To further clarify this point, at the end of the Introduction we now state "These results demonstrate that this self-inoculation treatment provides greater protection and consolidation effectiveness for salt weathered stones than the previously proposed bio-treatments tested here."

Some other comments are:

Lines 63-65: It is mentioned that exogenous bacteria are likely to be at a competitive disadvantage. But even if bacteria are isolated from the stone itself and one strain is stimulated and applied again, will this stimulated strain not also be pushed back again by the other bacteria present to move towards the original equilibrium? Would there be really a difference with exogenous bacteria (obtained from another source but in reasonably similar conditions)?

The reviewer is correct in pointing out that a stone-isolated single bacterial culture would face similar issues to an exogenous culture in terms of being at a competitive disadvantage. This has now been noted in the Ms.

Note, however, that our Type 3 treatment involved the application of the whole carbonatogenic bacterial community isolated from the stone to be treated. Therefore, such a bacterial community, in principle, would not alter the equilibrium of the stone carbonatogenic microbial community once applied to the stone substrate. This comment shows that indeed we were not fully successful when explaining how the treatment was prepared and applied (as pointed out by the other two referees). This is now solved by our detailed explanation of how this type 3 treatment was performed (see our answer to referees #1 and #2).

Lines 101-103: the differences in DR between weathered and unweathered stones are really small (in comparison to the large standard deviation). Are these differences significant? The same remark can be made for Fig 5.

DR was actually 4 times higher in treated stone as compared with the untreated sample. An important increase, as has also been recognized by referee #1. Our results (Fig. 5) further show that the consolidation level reached after treatment with the indigenous carbonatogenic bacteria (Fig. 5a and 5d) is significant and well above the standard deviation. Note that the calcarenite is a heterogeneous natural stone, and therefore displays some DR variations (reflected here by the values of standard deviation). This is why in order to have statistically significant results, we performed several DR measurements per treated and control areas (as stated in the Methods section).

Fig 6. Do the XRD peaks point at more quartz after treatment? Why would this be?

This is just due to the heterogeneous nature of the stone, which may lead to minor (few %) variations in the quartz content among different samples in the same stone block. In any case, the quartz content is always equal or below 5 wt %.

Line 291. The “f” after UFC/g can be omitted?

We apologize for the mistake. The correct spelling of Colony Forming Units is CFU. This error has been corrected throughout and the “f” was deleted.

Line 355. If there is no pore plugging at all (suppl fig 6 even points at pore coarsening?), is it not expected that the salts will easily penetrate to the unprotected zone underneath the outer few mm of the stone?

The salts come from the ground and reach the stone surface/subsurface at the stone block wall via capillary rise (flow through the stone block pore system). This is the most common and damaging situation in stone buildings subjected to salt weathering (see ref. #26: Schiro et al., Phys. Rev. Lett. 109, 265503, 2012). The presence of coarser pores would not lead to an increase in penetration of salts from the surface into the interior of the stone. Rather, a pore coarsening would encourage the saline solution to more easily reach the stone surface prior to crystallization, favoring formation of non-damaging salt efflorescence. In any case, our MIP results (Supplementary Fig. 6) do not show pore coarsening after treatment. Rather they show a reduction in the main pore size from ~10 μm to ~5 μm.

Line 356. How was it proven that CaCO₃ was “abundant”?

Our SEM observations show massive presence of bacterial carbonate. Moreover, bacterial carbonates were responsible for the nearly 10% reduction in the total porosity of the stone (MIP results). These results prove that abundant bacterial calcium carbonate was formed. To clarify this point we now state: "*The overall ~10% reduction in porosity further confirms that abundant bacterial calcium carbonate was formed.*"

Complete discussion part: some paragraphs could probably be removed since there is no additional information that is not already stated in the results part (e.g. check lines 397-401; 407-409; 361-371)

Thanks for the suggestion. We have removed those paragraphs, except lines 361-371, which we believe are necessary to put in context our results (i.e., comparison between the three treatments).

Line 561-566: had all stones similar humidity at the moment of the tape test? The humidity could affect the adhesion of the tape.

We fully agree with this statement. To avoid variations in surface adhesion due to humidity variations, this test was performed in spring and early fall when the relative humidity in Granada is typically lower than 50 % (dry weather). No peeling or DR tests were performed after/during rainy periods (which are rather rare in Granada). We state so in the Methods section.

REVIEWERS' COMMENTS:

Reviewer #1 (Remarks to the Author):

The authors have responded satisfactorily to the questions raised by the reviewers. In my opinion, the paper is ready for publication.

Reviewer #3 (Remarks to the Author):

It seems that the authors have taken great effort to address all reviewers' comments and therefore according to me the paper can now be accepted for publication.

Reviewer #4 (Remarks to the Author):

The manuscript NCOMMS-16-25358A, which deals with the development of an environmentally friendly, bacterial self-inoculation approach for the conservation of stone, is well written, technically sound, and it represents a new contribution in the field.

All the Reviewer's comments and suggestions have been addressed satisfactorily, and I think the manuscript has been greatly improved by these revisions.

My only recommendation is to support data (supplementary table 1 and Figure 8) with a simple statistical analysis, like t-test or ANOVA followed by post-hoc analysis, in order to find statistically significant differences among samples.

In figure 8, the differences among treatments are pretty evident, but I cannot say the same for data on supplementary table 1. The statistical analysis of variance could improve the interpretation of the results.

Response to Reviewers

Reviewer #1 (Remarks to the Author):

The authors have responded satisfactorily to the questions raised by the reviewers. In my opinion, the paper is ready for publication.

A: We thank again this referee for his/her comments and suggestions which helped us to improve the overall quality of our manuscript.

Reviewer #3 (Remarks to the Author):

It seems that the authors have taken great effort to address all reviewers' comments and therefore according to me the paper can now be accepted for publication.

A: We thank again this referee for his/her comments and suggestions which helped us to improve the overall quality of our manuscript.

Reviewer #4 (Remarks to the Author):

The manuscript NCOMMS-16-25358A, which deals with the development of an environmentally friendly, bacterial self-inoculation approach for the conservation of stone, is well written, technically sound, and it represents a new contribution in the field.

All the Reviewer's comments and suggestions have been addressed satisfactorily, and I think the manuscript has been greatly improved by these revisions.

A: We thank the referee for the overall positive evaluation of our study.

My only recommendation is to support data (supplementary table 1 and Figure 8) with a simple statistical analysis, like t-test or ANOVA followed by post-hoc analysis, in order to find statistically significant differences among samples.

In figure 8, the differences among treatments are pretty evident, but I cannot say the same for data on supplementary table 1. The statistical analysis of variance could improve the interpretation of the results.

A: We agree with this comment: While in the case of data presented in Figure 8, the differences among treatments are evident, this is not the case for the color variations presented in Supplementary Table 1. Therefore, following the suggestion of this referee we performed a statistical analysis (*t*-test) of these results.

In the corresponding parts of the Methods section we now state: "*The statistical significance of color measurements (before and after treatment) was evaluated by using the t-test implemented in the Statgraphics Centurion XVI.II code (Statpoint Technologies, Inc. The Plains, Virginia, USA).*" and "*T-tests were implemented to evaluate the statistical significance of the changes in dissolution rate taking place after bio-treatment.*" Similarly, in the corresponding parts of the Results section we now state for the case of *M. xanthus* and M-3P treatments that "*Chromatic changes (ΔE values) (Supplementary Table 1) over the 24 month study period were ≤ 3.6 . Moreover, t-tests showed that p-values were higher than 0.05, with the exception of the stone treated with M-3P (after 12 months) where the p-value was slightly below 0.05 (Supplementary Table 1). These results show that in nearly all cases the null hypothesis, namely that the difference between the mean ΔE values of the control and treated stones is zero, could not be rejected. This means that color changes were not statistically significant (95% confidence interval) in most cases, and in the only case where color changes were statistically significant, the ΔE value was still below the generally acceptable maximum $\Delta E=5.29$ ", and for the case of the bacterial self-inoculation treatment "*In this case, t-tests showed that p-values were higher than 0.05, with the exception of color changes after 24 months, where the p-value was below 0.05 (Supplementary Table 1). Nonetheless, even in this latter case where the color change was statistically significant, the ΔE value was below 5.29*". Note that p-values have been included in Supplementary Table 1. In the case of the dissolution experiments, we now state: "*The differences (reduction) in dissolution rate after bacterial treatment were statistically significant as demonstrated by t-tests showing p-values < 0.05 (0.00005 and 0.000002 for H_2O and 1M $MgSO_4$ dissolution experiments, respectively).*"*